# HarmonyGNNs: Harmonizing Heterophily and Homophily in GNNs via Self-Supervised Node Encoding

**Rui Xue & Tianfu Wu**
Department of Electrical and Computer Engineering (ECE)
North Carolina State University, Raleigh, NC 27695, USA
`{rxue,twu19}@ncsu.edu`

## Abstract

Graph Neural Networks (GNNs) have made significant advances in representation learning on various types of graph-structured data. However, GNNs struggle to simultaneously model heterophily and homophily, a challenge that is amplified under self-supervised learning (SSL) where no labels are available to guide the training process. This paper presents **HarmonyGNNs** , an end-to-end graph SSL framework designed to **harmonize heterophily and homophily** through two complementary innovative perspectives: **(i) Representation Harmonization via Joint Structural Node Encoding.** Nodes are embedded into a unified latent space that retains both node specificity and graph structural awareness for harmonizing heterophily and homophily. Node specificity is learned via linear and non-linear node feature projections. Graph structural awareness is learned via a proposed Weighted Graph Convolutional Network (WGCN). A self-attention module enables the model learning-to-adapt to varying levels of patterns. **(ii) Objective Harmonization via Predictive Architecture with Node-Difficulty–Aware Masking.** A teacher network processes the full graph. A student network receives a partially masked graph. The student is trained end-to-end, while the teacher is an exponential moving average of the student. The proxy task is to train the student to predict the teacher's embeddings for all nodes (masked and unmasked). To keep the objective informative across the graph, two masking strategies that guide selection toward currently hard nodes while retaining exploration are proposed. **Theoretical underpinnings of HarmonyGNNs** are also analyzed in detail. Comprehensive evaluations on benchmarks demonstrate that HarmonyGNNs achieves state-of-the-art performance on heterophilic graphs (e.g., +7.1% on Texas, +9.6% on Roman-Empire over the prior art) while matching SOTA on homophilic graphs, and delivering strong computational efficiency. Code is released at this Github repository.

## 1 Introduction

Representation learning on graph-structured data has emerged as a vibrant research area, serving as a cornerstone for a wide range of graph learning tasks, including node classification, link prediction, and graph classification (Kipf & Welling, 2016a; Gasteiger et al., 2019; Veličković et al., 2017; Wu et al., 2019). These tasks are critical in diverse real-world domains such as recommendation systems, molecular biology, and transportation (Tang et al., 2020; Sankar et al., 2021; Fout et al., 2017; Wu et al., 2022; Zhang et al., 2024). Graph Neural Networks (GNNs) have become the dominant paradigm for learning expressive node and graph representations (Hamilton, 2020; Gasteiger et al., 2018; Veličković et al., 2017).

Traditional GNNs are typically trained in a semi-supervised manner and have demonstrated impressive performance across numerous benchmarks (Xu et al., 2018; Li et al., 2021; Sun et al., 2021; Xue et al., 2023a; 2024). However, these semi-supervised methods heavily rely on the availability of labeled data, making them vulnerable to significant performance degradation when labeled data is scarce (Xue et al., 2023b). To overcome the limitations of label scarcity, Self-Supervised Learning (SSL)

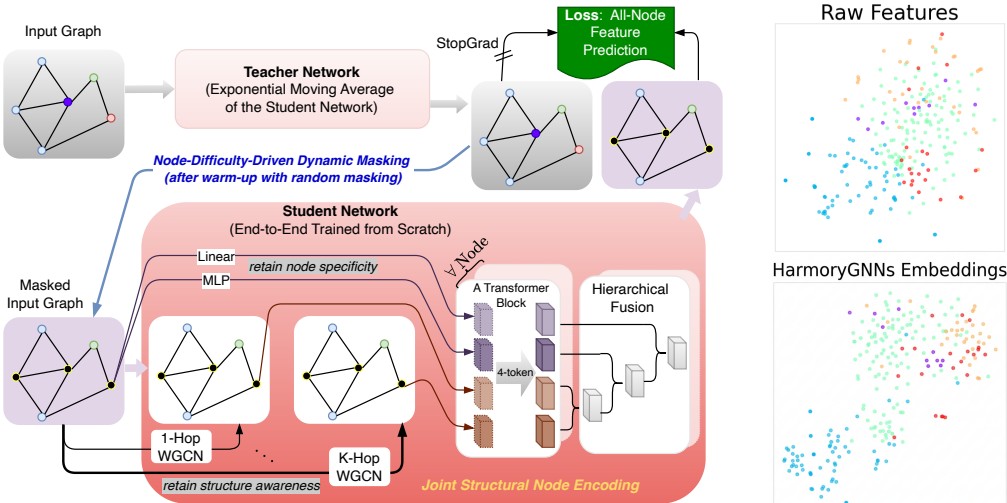

Figure 1: Illustration of our proposed HarmonyGNNs .

Figure 2: T-SNE on Wisconsin.

has emerged as a promising alternative. Various graph SSL methods (Velickovic et al., 2019; Zhu et al., 2020b; Hou et al., 2022; Chen et al., 2022; Xiao et al., 2024; Tang et al., 2022; Xiao et al., 2022; Yuan et al., 2023) have demonstrated strong performance under low-label regimes. However, current SSL paradigms—whether contrastive or generative—suffer from their own drawbacks. Contrastive methods often rely on complex training pipelines and carefully crafted data augmentations, while generative methods are prone to reconstruction-space mismatches. A more comprehensive review of related work is provided in Appendix A.

More importantly, real-world graphs exhibit complex mixed structural patterns, where homophily (the tendency of connected nodes to share similar labels) and heterophily (the presence of dissimilar labels among connected nodes) coexist at both local and global scales. We provide a visualization in Fig. 2. And intensities are varying across datasets. For example, the Roman-Empire dataset exhibits a homophily ratio of only 0.05, while Cora shows a ratio of 0.81 (see Table 1 for details). Many existing graph SSL models still perform poorly on heterophilic graphs, undermining their generalization capabilities. This is particularly troubling given SSL's fundamental reliance on raw graph structure and node features without explicit label guidance.

Recent efforts have attempted to address this challenge. Methods such as MUSE (Yuan et al., 2023), GREET (Liu et al., 2022), and GraphACL (Xiao et al., 2024) have shown promise in improving SSL performance on heterophilic graphs. However, achieving robust performance across both homophilic and heterophilic patterns remains elusive. This persistent challenge stems from a deeper issue: the inability of current graph SSL frameworks to harmonize the mixed structural patterns.

We propose that harmonizing homophily and heterophily within a single graph SSL framework is key. Specifically, a unified model should achieve both **objective harmonization** and **representation harmonization** when handling mixed structural patterns. Regarding objective harmonization, selecting an appropriate proxy task is crucial. Contrastive approaches in SSL rely on relative objectives (e.g., InfoNCE) without a stable global reference, making it unclear which pattern should dominate in mixed graphs. This region-dependent ambiguity prevents convergence to a unified latent space; Generative methods that force raw feature reconstruction yield contradictory signals when neighbors have dissimilar attributes in heterophilic settings. In terms of representation harmonization, homophilic regions require smoothness to capture similarity, while heterophilic regions demand distinctiveness to preserve differences. Existing methods cannot adaptively balance these needs, and thus are biased toward one structural pattern.

To this end, we present HarmonyGNNs (Fig. 1), an end-to-end graph SSL framework that achieves both objective and representation harmonization. We summarize our contributions as follows:

- **Objective Harmonization via Predictive Architecture with Dynamic Masking:** We exploit a Teacher-Student framework which provides stable, holistic guidance in Graph SSL. The teacher, with a full view of the unmasked graph, produces holistic node representations as node-encoding

anchors, capturing both homophilic and heterophilic relations. The student is then guided to predict this stable target. Crucially, the teacher's EMA-updated parameters ensure the learning spaces are aligned and prevent the student from being misled by noisy, oscillating updates, which is critical for adapting to complex structures. Due to the interconnected nature of graphs, we compute the prediction loss for the entire graph (rather than only the masked nodes), thereby addressing the severer ambiguity inherent in graph data. Furthermore, instead of random node masking, we propose two dynamic masking strategies, which generate training tasks that are both challenging and informative. This design yields a learning objective that harmonizes easy and hard samples as well as homophilic and heterophilic signals.

- **Representation Harmonization via Joint Structural Node Encoding:** To enhance representation learning, we combine linear and MLP-based node feature transformations (emphasizing intrinsic attributes) with K-hop structural projections via proposed Weighted GCN (which adaptively aggregates neighbor information). A vanilla Transformer block integrates these representations via self-attention, ensuring adaptability to homophily and heterophily while maintaining efficiency. A novel hierarchical fusion strategy is applied to integrate/calibrate the different types of representations. It gives the model the ability to "see" and learn different patterns.

These two components are fundamentally intertwined, and each of them is essential. The predictive architecture provides the learning stability, the joint encoding module provides the expressive power to handle mixed signals (see an illustration in Fig. 2), and the dynamic masking strategy provides a challenging yet meaningful learning objective. Extensive experiments on various mixed-structure graph benchmark datasets verify the strong performance of our HarmonyGNNs , demonstrating improved training effectiveness, efficiency, and generalization. The results show that a single, unified framework can be designed to automatically navigate the full homophily-heterophily spectrum without requiring any prior knowledge of the graph's properties.

## 2 PRELIMINARY

We present a preliminary analysis demonstrating the inability of baseline methods to effectively learn homophily and heterophily mixed patterns, which motivates our proposed HarmonyGNNs .

**Notation 1.** *Denote by $G = (V, E)$, a graph with the node set $V$ of $N$ nodes and the edge set $E$. Each node $v \in V$ has a $d$-dim feature vector $f(v) \in \mathbb{R}^d$. A subset $\mathbf{V} \subseteq V$ carries labels $\ell(v) \in \mathcal{Y}$, these labels are not used during self-supervised training and used only for linear probing and $k$-means evaluation with self-supervised node encoding frozen.*

**Homophily and Heterophily in Graphs.** In graphs, homophily means that adjacent nodes $(u, v)$ tend to have similar features, and heterophily means the opposite, which can be reflected in the graph normalized Laplacian quadratic form, $f^\top \cdot L_{sym} \cdot f = \sum_{(u,v) \in E} A_{uv} \left( \frac{f(u)}{\sqrt{d_u}} - \frac{f(v)}{\sqrt{d_v}} \right)^2$, where $L_{sym}$ represents the symmetric normalized Laplacian, $L_{sym} = \mathbb{I} - D^{-\frac{1}{2}} \cdot A \cdot D^{-\frac{1}{2}}$ with the degree matrix $D$, adjacency matrix $A$, and an identity matrix $\mathbb{I}$. $d_u$ and $d_v$ are the node degrees. In a homophilic graph, $f(u) \approx f(v)$ for adjacent nodes, making $f^\top \cdot L_{sym} \cdot f$ small. Conversely, in heterophilic graphs, the differences $\left( \frac{f(u)}{\sqrt{d_u}} - \frac{f(v)}{\sqrt{d_v}} \right)^2$ are larger. The coexistence of homophiliy

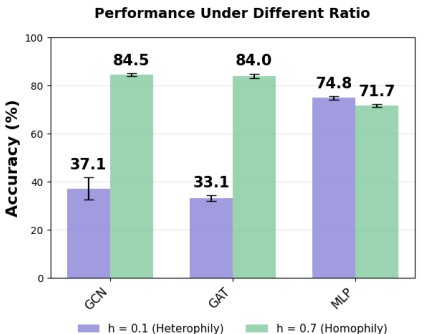

Figure 3: Impacts of homophily ratios.

and heterophily in real-world graph data challenges representation learning, especially via graph SSL.

**Control Experiments using Synthetic Graphs.** To illustrate the impacts of varying homophily ratios in graph data, we leverage synthetic graphs (Zhu et al., 2020a) with controlled homophily ratios, $h$ ($h = 0.1$ indicates strong heterophily and $h = 0.7$ corresponds to homophily) in training GNNs under supervised learning setting. We train classic GCN and GAT, and a simple baseline node-based MLP (with graph structure not used) which is found useful in (Chen et al., 2022).

Fig. 3 shows the results. As expected, GCN and GAT show much stronger performance on homophilic graphs than heterophilic ones. The baseline MLP significantly improves performance on heterophilic

graphs, thanks to its capability of retaining node specificity, at the expense of degrading performance on homophilic graphs (due to lacking graph structural awareness). So, we can clearly see the advantage of adaptively harnessing the strength of node-specificity representation and graph structural awareness, which motivates our HarmonyGNNs .

# 3 METHOD

We first present details of *Objective Harmonization* and *Representation Harmonization* in our HarmonyGNNs in Sec. 3.1 and 3.2 respectively, followed by theoretical underpinnings comparing our HarmonyGNNs to existing methods to highlight the strengths of our HarmonyGNNs in Sec. 3.4.

## 3.1 OBJECTIVE HARMONIZATION

The choice of proxy task in SSL is critical. Inappropriate tasks can actually degrade performance (see details in Appendix A). In our HarmonyGNNs , motivated by JEPA and its demonstrated success in computer vision (Assran et al., 2023), we employ **a teacher–student predictive architecture** to eliminate the need for complex negative sampling and to prevent representation collapsing, while ensuring both feature prediction in an aligned latent space and stable representation learning (see Sec 3.4). Moreover, we adopt **masked node modeling** as our primary proxy task and introduce two novel **node-difficulty-driven dynamic masking** strategies that encourage the model to learn more robust and generalizable representations.

**Masked Node Modeling with Teacher-Student Predictive Architecture.** For an input graph $G = (V, E)$ and a given node-wise mask $\mathcal{M}$, let $V_m = \{ v \in V \mid \mathcal{M}(v) = 1 \}$ be the subset of masked nodes, and $V_u = V \setminus V_m$ the subset of remaining unmasked nodes. For the masked nodes in $V_m$, we replace their raw input features by learnable parameters with random initialization, e.g., from the white noise distribution, $f(v) \sim \mathcal{N}(0, 1), \forall v \in V_m$. Let $\mathbb{G} = (V_u \cup V_m, E)$ denote the partially masked input graph, which is generated at each training iteration by sampling a node-wise mask $\mathcal{M}$. To facilitate learning a proper latent space, we leverage **a teacher-student predictive architecture**. Denote the student and the teacher network by $S(\cdot; \Phi)$ and $T(\cdot; \Psi)$, parameterized by $\Phi$ and $\Psi$ respectively. The student network sees the masked input graph $\mathbb{G}$, while the teacher network sees the full graph $G$. The teacher network has the exactly same network configuration as the student, and is not trained, but uses the exponential moving average (EMA) of the student network to ensure the stability of training and the convergence of the same latent space (He et al., 2020; Assran et al., 2023; Bardes et al., 2024):

$$\Psi_i = \alpha \cdot \Psi_{i-1} + (1 - \alpha) \cdot \Phi_i, \quad i = 1, 2, \cdots, I \tag{1}$$

**All-Node Feature Prediction in the Latent Space.** To estimate the student network's parameters $\Phi$, a proxy or pretext task is entailed. One common approach is to consider masked nodes feature prediction in $V_m$ only. However, graph nodes are inherently more ambiguous because their interconnections create strong dependencies, leading to interactions between masked and unmasked nodes to be captured. Predicting only masked nodes' features between the student and the teacher network is thus suboptimal for learning a more meaningful latent space. For a node $v \in V = V_m \cup V_u$, denote the outputs from the student and teacher network by $S(v; \Phi) \in \mathbb{R}^D$ and $T(v; \Psi) \in \mathbb{R}^D$ respectively. We propose to compute the prediction loss in the latent space based on the entire graph,

$$\mathcal{L}(\Phi) = \frac{1}{N} \sum_{v \in V} ||S(v; \Phi) - T(v; \Psi)||_2^2. \tag{2}$$

**Node-Difficulty-Driven Dynamic Node Masking.** Masking strategies are critical for the success of SSL. In general, random masking with sufficient high masking ratios (Devlin et al., 2019; He et al., 2022) leads to hard proxy tasks to be solved via learning meaningful representations. However, given the complex and often unknown topological properties of graphs, random masking alone is insufficient to guide effective SSL. Hence, we propose two novel dynamic masking strategies to compute the mask $\mathcal{M}_i$ at each iteration. These strategies adaptively consider each node's learning difficulty based on the prediction loss in Eqn. 2, ensuring that the prediction task is sufficiently challenging to learn robust representations with excellent generalization capabilities.

Denote by $R$ be the overall node masking ratio hyperparameter ($R \in (0, 1)$). We mask $M = \lfloor N \times R \rfloor = |V_m|$ nodes in total. We warm up the training with purely random masking for a

predefined number of epochs. Afterwards, we adopt the exploitation-exploration strategy, where we exploit two node-difficulty-driven dynamic masking approaches, combined with the purely random exploration-based masking. Let $r$ be the exploitation ratio ($r \in [0, 1]$), we first select $m = \lfloor M \times r \rfloor$ nodes using the exploitation approach, and the remaining $M - m$ nodes are randomly sampled from the set of available $N - m$ nodes (without replacement).

• **HarmonyGNNs +Diffi: Node Feature Prediction Loss Driven Masking.** Based on Eqn. 2, we define the difficulty score of a node $v$ after the current iteration by,

$$\text{Diffi}(v) = ||S(v) - T(v)||_2^2, \tag{3}$$

which is used to compute the mask for the next iteration. We sort the nodes $v \in V$ based on $\text{Diffi}(v)$ in a decreasing order, and then select the first $m$ nodes to mask. This approach ensures that the model focuses on nodes where the student network's understanding is significantly lacking compared to the teacher network, thereby driving the student network to improve its representations where it is most deficient. However, this approach does not entirely prevent the issue of over-focusing on a small subset of high-difficulty nodes while neglecting the overall data diversity. To address this, we seek a probabilistic solution in the next approach.

• **HarmonyGNNs +Prob: Masking via Bernoulli Sampling with Node-Difficulty Informed Success Rate.** Let $p_v$ be the success rate of the Bernoulli distribution used for selecting the node $v \in V$ to be masked, i.e., $\mathcal{M}(v) \sim \text{Bernoulli}(p_v)$. We have,

$$p_v = p_0 + \delta_v, \quad p_0 = (1 - r) \times R, \quad \delta_v = \left( \frac{\text{Diffi}(v)}{\text{Diffi}_{\max}} \right) \times r \times R, \tag{4}$$

where $p_0$ is the base success rate subject to the exploration approach, and it is the same for all nodes. $\delta_v$ is the node-difficulty based exploitation with $\text{Diffi}_{\max}$ the maximum value of the node difficulty score among all nodes. This approach ensures that all nodes have a base probability $p_0$ of being masked, while higher-difficulty nodes are masked with a greater chance, effectively guiding the model to focus more on learning from these challenging nodes. Since this approach is a node-wise Bernoulli sampling, to prevent the worst cases in which either too few nodes or too many nodes (much greater than $M$) are actually masked, we do sanity check in the sampling process by either repeatedly sampling (if too few nodes have been masked) or early stopping.

## 3.2 Representation Harmonization

With the above architectural designs and loss function choices, we seek node encoding scheme towards the expressivity of node features in graph SSL in terms of inducing heterophily and homophily awareness and adaptivity in $S(v)$ against the raw input features $f(v)$ for downstream tasks.

**Learning Weighted GCN for Heterophily-Preserved Homophily Awareness**. The traditional GCN has been proven to act as a simple and efficient smoothing operator (Kipf & Welling, 2016a) , making it good for homophilic graphs, but becoming less effective for heterophilic graphs (see Fig. 3). To address this, we introduce Weighted GCN (WGCN), which learns weights for edges and thus adaptively controls message passing—balancing smoothing and sharpening—to handle diverse graph structures more effectively, avoid complex design choices and preserve high efficiency. Formally, a WGCN's layer is given by,

$$H^{(l+1)} = \sigma(\mathcal{A} \cdot H^{(l)} \cdot W^{(l)}), \tag{5}$$

where $\mathcal{A}_{ij}$ is a learnable parameter that adjusts the edge weight dynamically, meaning the model learns how much influence each neighbor should have, instead of treating all edges equally. It is initialized from $\tilde{A} = \tilde{D}^{-1/2}(A + \mathbb{I})\tilde{D}^{-1/2}$, the normalized adjacency matrix with self-loops. Then, the learned edge weight are passed through a Sigmoid function $a_e = \sigma(w_e)$ as the edge weight for edge $e$. Bounding $a_e$ to $(0, 1)$ regularizes the magnitude of edge weights, mitigating the trivial global-rescaling ambiguity that can otherwise arise in weighted message passing. $H^{(l)} \in \mathbb{R}^{N \times C}$ is the node feature matrix at layer $l$ with the output dimension $C$ is chosen to control model complexity. $W^{(l)}$ is the trainable weight matrix. In homophilic regions, WGCN retains high weights for similar neighbors; in heterophilic regions, it downweights dissimilar ones, preventing oversmoothing and capturing complex structures more effectively.

**Projecting Node-Wise Features for Heterophily-Targeted Awareness**. From Fig. 3, we can see the base MLP can retain node specificity for achieving good performance on heterophilic graphs. So,

we introduce a nonlinear projection $f^{(Mlp)}(v)$ on the node features. Additionally, the node features themselves play crucial roles, especially when neighborhoods exhibit high heterophily (Yuan et al., 2023). Hence, we also apply a linear projection $f^{(Linear)}(v)$.

**Learning Multi-Head Self-Attention for Heterophily and Homophily Adaptivity.** To adaptively capture both homophily and heterophily, for a node $v \in V$, we map it into a joint latent space. For example, we can simply combine the four types of features,

$$\mathbf{f}(v) = \left[ f^{(Linear)}(v) \oplus f^{(Mlp)}(v) \oplus H^{(\ell)}(v) \oplus H^{(\ell')}(v) \right], \quad where \quad \mathbf{f}(v) \in \mathbb{R}^{4 \times C} \quad (6)$$

where $\cdot \oplus \cdot$ denotes stacking operation, $\ell$ and $\ell'$ denote WGCN layers, which can be tuned easily. To mix and re-calibrate the different types of features per node to induce heterophily and homophily awareness and adaptivity, we treat the each projection output as a "token" (e.g., 4 tokens as illustrated in Fig. 1), and apply a vanilla Transformer block (Vaswani et al., 2017) with pre-norm settings. By doing so, we maintain the efficiency with our novel **feature level** attention mechanism, which is different from existing graph transformer works that aim to capture node-wise attention and suffer from scalability caused by the quadratic complexity of the Transformer model w.r.t. the number of nodes $N$.

**Fusing and Selecting Tokens Hierarchically as SSL Node Encoding.** The four tokens in Eqn. 6, after passing through Transformer block, provide complementary representations of each node. Instead of flattening them all at once, we fuse the most closely related encoded tokens first and propagate the result upward, which (i) keeps the parameter count low, (ii) eases gradient flow, and (iii) lets the model learn a coarse-to-fine weighting of homophilic and heterophilic patterns. We first fuse the two encoded tokens generated by WGCN; we then iteratively merge this result with each of the remaining two encoded projection tokens to produce the final output.

$$S(v) = \sigma \left( \text{Linear} \left( X_{0,C} || \sigma \left( \text{Linear} \left( X_{1,C} || \sigma \left( \text{Linear} \left( X_{2,C} || X_{3,C} \right) \right) \right) \right) \right) \right), \quad S(v) \in \mathbb{R}^C, \quad (7)$$

where $X_{i,C}$ represents the output of $f^{(Linear)}$, $f^{(Mlp)}$, $H^{(l)}$ and $H^{(l')}$ from the Transformer block for $i = 0, 1, 2, 3$ respectively. Additionally, we also offer several strategies for deriving the final output, such as taking the mean, the max, and simply selecting $X_{0,C}$. We provide an ablation study about the encoded token selection in Appendix M.

### 3.3 COMPLEXITY ANALYSIS

Let $h$ denote the hidden (channel) dimension, $l$ the number of WGCN layers and $s$ the number of per-node tokens used in the harmonization module (e.g., $s = 4$ in our implementation).

**Computational complexity.** The computation of HarmonyGNNs can be decomposed into three parts:

- MLP branch: Mapping raw node features to node-specific tokens via one or two linear/MLP layers costs $\mathcal{O}(Ndh)$ or $\mathcal{O}(Ndh + Nh^2)$ (for a 2-layer MLP).
- WGCN branch: The $l$-hop structural embeddings are implemented with $l$ sparse WGCN layers, each with complexity $\mathcal{O}(|E|h)$. Hence the cost is $\mathcal{O}(l|E|h)$.
- Feature-level MHSA: For each node, we apply MHSA over its $s$ feature tokens. Computing attention scores between $s$ tokens and the associated linear projections costs $\mathcal{O}(s^2h + sh^2)$ per node, and thus $\mathcal{O}(Ns^2h + Nsh^2)$ over all nodes. **Note that the cost is not $\mathcal{O}(N^2h)$. This is exactly why we emphasize feature-token attention instead of node-wise attention.**

Putting everything together, the overall complexity of the student encoder is $\mathcal{O}\big(Ndh + l|E|h + Ns^2h + Nsh^2\big)$. Since $l$ and $s$ are small constants (e.g., $l \leq 2$ and $s = 4$) in our implementation, this simplifies to $\mathcal{O}\big(Ndh + |E|h + Nh^2\big)$, which is on the same order as standard GNN encoders and strictly more efficient than node-wise graph transformers with $\mathcal{O}(N^2h)$ attention.

**Memory complexity.** The memory footprint of the proposed method is dominated by:

- Node activations: Each node stores $s$ tokens of dimension $h$, leading to $\mathcal{O}(Nsh) = \mathcal{O}(Nh)$
- Graph structure: The sparse edge index and a scalar weight per edge in WGCN require $\mathcal{O}(|E|)$
- Model parameters: Input projections, WGCN weights, and token-level attention/MLPs take $\mathcal{O}(dh + h^2)$, which is independent of $N$ and $|E|$.

Under the sparse-graph assumption, the total memory complexity is therefore $\mathcal{O}(Nh+|E|+dh+h^2)$, i.e., it grows **linearly** in both $N$ and $|E|$ and does not require storing any $\mathcal{O}(N^2)$ node-wise attention maps.

## 3.4 Theoretical Underpinnings

In this section, we provide theoretical underpinnings of graph SSL convergence analyses for our HarmonyGNNs and alternative encoder-decoder based graph SSL methods such as GraphMAE (Hou et al., 2022; 2023) (which aim to directly reconstruct raw input features of masked nodes).

The encoder-decoder SSL architecture consists of an encoder network $E(\cdot;\Theta_{enc})$ and a separate decoder network $D(\cdot;\Theta_{dec})$. Let $\theta = (\Theta_{enc}, \Theta_{dec})$ collects all parameters. Given a masked graph signal $\bar{f}$ from the input graph signal $f$ of $N$ nodes using a mask $\mathcal{M}$, its objective is to minimize,

$$\mathcal{L}_{E-D}(\theta) = \frac{1}{N}\|D\big(E(\bar{f};\Theta_{enc});\Theta_{dec}\big) - f\|_2^2. \tag{8}$$

The convergence rates of encoder-decoder methods and our HarmonyGNNs (Eqn. 2) can be bounded in the main theorem as follows.

**Theorem 1.** *Consider the optimization of encoder-decoder based graph SSL in Eqn. 8 and our proposed HarmonyGNNs in Eqn. 2 under the same encoder architecture and following assumptions/conditions:* (i) Smoothness & Lipschitz: *The encoder $E(\cdot;\Theta_{enc})$ and decoder $D(\cdot;\Theta_{dec})$ are $\beta$-smooth and $L$-Lipschitz;* (ii) Boundedness: *Gradients of the encoder $\|\nabla E(\cdot;\Theta_{enc}^{(t)})\|$, gradients of the decoder $\|\nabla D\big(E(\cdot;\Theta_{enc}^{(t)});\Theta_{dec}^{(t)}\big)\|$, and reconstruction errors $\|D\big(E(\bar{f};\Theta_{enc}^{(t)});\Theta_{dec}^{(t)}\big) - f\|$ are bounded;* (iii) Strong convexity: *Both the encoder $E(\cdot;\Theta_{enc})$ and decoder $D(\cdot;\Theta_{dec})$ are $\mu$-strongly convex in their parameters;* (iv) Approximation: *With only unmasked inputs, the encoder–decoder (or teacher–student in HarmonyGNNs) incurs approximation error $\epsilon_{E-D}$ (or $\epsilon_{T-S}$). See assumptions details in Appendix G. Then, the following three results hold:*

- **A. Linear Convergence Bounds Under Strong Convexity.** *For our HarmonyGNNs,*

$$\|\Phi^{(t+1)} - \Phi^*\|^2 \leq (1 - \frac{\mu_E^2}{\beta_E^2}) \cdot \|\Phi^{(t)} - \Phi^*\|^2 \tag{9}$$

  *For the encoder-decoder models,*

$$\|\theta^{(t+1)} - \theta^*\|^2 \leq \left(1 - \frac{\min(\mu_E^2, \mu_D^2)}{\max(\beta_E^2, \beta_D^2)}\right) \cdot \|\theta^{(t)} - \theta^*\|^2 \tag{10}$$

  *from which we can see our HarmonyGNNs converges to the optimal solution $\Phi^*$ faster than the encoder-decoder counterpart to their optimal solutions $\Theta^*$ due to a smaller contraction factor $\left(1 - \frac{\mu_E^2}{\beta_E^2}\right) < \left(1 - \frac{\min(\mu_E^2, \mu_D^2)}{\max(\beta_E^2, \beta_D^2)}\right)$. This implies that HarmonyGNNs can achieve a faster convergence.*

- **B. Proxy Task Loss Bounds** *under a Lipschitz-dependent assumption between the masked graph signal and the raw graph signal, $\|\bar{f} - f\| \leq \delta$. For our HarmonyGNNs,*

$$\|S(\bar{f};\Phi) - T(f;\Psi)\| \leq L_E \cdot \delta + \epsilon_{T-S}. \tag{11}$$

  *For the encoder-decoder models,*

$$\|D\big(E(\bar{f};\Phi_{enc});\Theta_{dec}\big) - f\| \leq L_E \cdot L_D \cdot \delta + \epsilon_{E-D}. \tag{12}$$

  *W.L.O.G., assume $\epsilon_{E-D} = \epsilon_{T-S}$, our HarmonyGNNs has a smaller error upper bound, indicating that our teacher–student model is closer to the optimal solution $\Phi^*$ during training, which implies that its parameter updates are more stable and its convergence speed is faster (as shown in **A**).*

- **C. Gradient-Difference Bounds** *in Encoder-Decoder Models Showing Coupling Effects of Parameter Updating,*

$$\|\nabla\mathcal{L}_{E-D}(\Theta_{enc}^{(t+1)}) - \nabla\mathcal{L}_{E-D}(\Theta_{enc}^{(t)})\| \leq 2B_{Reconst}\Big(\beta_E B_D + B_E L_D L_E\Big)\|\Theta_{enc}^{(t+1)} - \Theta_{enc}^{(t)}\|+$$

$$2B_E B_{Reconst}\beta_D\|\Theta_{dec}^{(t+1)} - \Theta_{dec}^{(t)}\| + 4B_E B_D B_{Reconst}, \tag{13}$$

$$\|\nabla\mathcal{L}_{E-D}(\Theta_{dec}^{(t+1)}) - \nabla\mathcal{L}_{E-D}(\Theta_{dec}^{(t)})\| \leq 2B_{Reconst}\beta_D L_E\|\Theta_{enc}^{(t+1)} - \Theta_{enc}^{(t)}\|+$$

$$2B_{Reconst}\beta_D\|\Theta_{dec}^{(t+1)} - \Theta_{dec}^{(t)}\| + 4B_D B_{Reconst}, \tag{14}$$

  *where the coupling effects in Encoder-Decoder models may lead to instability in learning. The proofs are provided in the Appendix G, H and I.*

Table 1: Results of node classification (in percent ± standard deviation across 10 splits). The best and the runner-up results are highlighted in red and blue respectively in terms of the mean accuracy.

| Methods / Datasets | Heterophilic | | | | Homophilic | | | |
|---|---|---|---|---|---|---|---|---|
| | Cornell | Texas | Wisconsin | Actor | Cora | CiteSeer | PubMed | Arxiv |
| Homo Ratio | 0.30 | 0.11 | 0.21 | 0.22 | 0.81 | 0.74 | 0.80 | 0.66 |
| DGI | 63.35±4.61 | 60.59±7.56 | 55.41±5.96 | 29.82±0.69 | 82.29±0.56 | 71.49±0.14 | 77.43±0.84 | 70.19±0.73 |
| GMI | 54.76±5.06 | 50.49±2.21 | 45.98±2.76 | 30.11±1.92 | 82.51±1.47 | 71.56±0.56 | 79.83±0.90 | 69.23±0.79 |
| MVGRL | 64.30±5.43 | 62.38±5.61 | 62.37±4.32 | 30.02±0.70 | 83.03±0.27 | 72.75±0.46 | 79.63±0.38 | 70.88±0.51 |
| BGRL | 57.30±5.51 | 59.19±5.85 | 52.35±4.12 | 29.86±0.43 | 81.08±0.17 | 71.59±0.42 | 79.97±0.36 | 71.24±0.35 |
| GRACE | 54.86±6.95 | 57.57±5.68 | 50.00±5.83 | 29.01±0.78 | 80.08±0.53 | 71.41±0.38 | 80.15±0.34 | 70.96±0.31 |
| GraphMAE | 61.93±4.59 | 67.80±3.37 | 58.25±4.87 | 31.48±0.56 | 84.20±0.40 | 73.20±0.39 | 81.10±0.34 | 71.75±0.17 |
| DSSL | 53.15±1.28 | 62.11±1.53 | 56.29±4.42 | 28.36±0.65 | 83.06±0.53 | 73.20±0.51 | 81.25±0.31 | 70.13±0.25 |
| NWR-GAE | 58.64±5.61 | 69.62±6.66 | 68.23±6.11 | 30.17±0.17 | 83.62±1.61 | 71.45±2.41 | 83.44±0.92 | 71.18±0.62 |
| HGRL | 77.62±3.25 | 77.69±2.42 | 77.51±4.03 | 36.66±0.35 | 80.66±0.43 | 68.56±1.10 | 80.35±0.58 | 68.55±0.38 |
| GraphACL | 59.33±1.48 | 71.08±2.34 | 69.22±5.69 | 30.03±1.03 | 84.20±0.31 | 73.63±0.22 | 82.02±0.15 | 71.72±0.26 |
| * S3GCL | 81.27±3.67 | 86.12±3.91 | 84.56±2.71 | 36.88±0.34 | *— | *— | *— | 71.36±0.60 |
| †MUSE | 82.00±3.42 | 83.98±2.81 | 88.24±3.19 | 36.15±1.21 | 82.22±0.21 | 71.14±0.40 | 82.90±0.40 | 70.98±0.32 |
| GREET | 73.51±3.15 | 83.80±2.91 | 82.94±5.69 | 35.79±1.04 | 83.84±0.71 | 73.25 ±1.14 | 80.29±1.00 | 71.09±0.43 |
| HarmonyGNNs +Diffi | 85.41±1.79 | 93.24±2.77 | 92.74±2.91 | 37.93±0.56 | 84.70±0.56 | 73.36±0.33 | 83.42±0.26 | 71.56±0.28 |
| HarmonyGNNs +Prob | 85.68±2.11 | 92.45±3.78 | 93.13±3.42 | 38.15±0.71 | 84.82±0.23 | 73.12±0.28 | 83.25±0.16 | 71.97±0.12 |

† MUSE only provides hyperparameters for Cornell in their official repo; however, their results were not reproducible based on the provided codes. And, no hyperparameters were provided for other datasets. We tried our best to tune its hyperparameters in comparisons.

⋆ S3GCL's official repo is under construction with codes to be factored and organized, so we directly report its published performance on all datasets except Cora, Citeseer, and Pubmed, for which different splits were used with higher label rates in linear probing.

## 4 EXPERIMENTS

**Datasets.** We evaluate our model on a suite of real-world benchmarks: **four widely adopted homophilic graphs** (Cora, CiteSeer, PubMed, and ArXiv) (Sen et al., 2008; Hu et al., 2021) and **seven heterophilic graphs** (including Cornell, Texas, Wisconsin, Actor, Chameleon, Squirrel and Roman-Empire) (Pei et al., 2020; Platonov et al., 2023). These datasets encompass various aspects and span both small-scale and large-scale networks, ensuring our experiments are diverse and comprehensive. Note that, as original Chameleon and Squirrel are known to be problematic (Platonov et al., 2023), we use their filtered versions to ensure an accurate assessment of model performance. We also provide the homo ratio in the table. Details of these datasets are summarized in Appendix O.

**Baselines.** To make fair comparisons with other baselines, we adopt the widely used node classification task as our main downstream evaluation. We also conduct the experiment of node clustering in Appendix D. Here, we primarily compare against two groups of SSL baselines (see Appendix C for semi-supervised comparisons): (1) **Traditional SSL methods**: DGI (Velickovic et al., 2019), GMI (Peng et al., 2020), MVGRL (Hassani & Khasahmadi, 2020), BGRL (Thakoor et al., 2021), GRACE (Zhu et al., 2020b), and GraphMAE (Hou et al., 2022); (2) **SSL methods tailored for heterophilic graphs**: DSSL (Xiao et al., 2022), NWR-GAE (Tang et al., 2022), HGRL (Chen et al., 2022), GraphACL (Xiao et al., 2024), S3GCL (Wan et al., 2024), GREET (Liu et al., 2022) and MUSE (Yuan et al., 2023). We also provide comparisons with additional baselines in Appendix F.

For evaluation, we follow the same protocol as all other baselines (Liu et al., 2022; Yuan et al., 2023) by freezing the trained SSL model and utilizing the generated embeddings for a downstream linear classifier. Note that we reproduce the results of major baselines (Liu et al., 2022; Hou et al., 2022; Xiao et al., 2024; Yuan et al., 2023) using the hyperparameters provided in their official repositories, and we ensure that the data split is consistent across all models. However, for those models among them that do not provide dataset-specific hyperparameters, such as MUSE, we conducted our own fine-tuning. For

Table 2: Results of node classification on three `heterophilic` graph datasets.

| Methods | Chameleon(filtered) | Squirrel(filtered) | Roman-Empire |
|---|---|---|---|
| Homo Ratio | 0.24 | 0.21 | 0.05 |
| DGI | 32.61±2.92 | 38.78±2.34 | 43.16±0.78 |
| BGRL | 32.55±4.65 | 35.67±1.42 | 52.16±0.25 |
| GRACE | 35.39±3.58 | 36.21±2.81 | 51.58±0.98 |
| MUSE | 46.48±2.51 | 41.57±1.44 | 66.26±0.53 |
| GREET | 44.67±2.98 | 39.69±1.85 | 63.37±1.91 |
| HarmonyGNNs +Diffi | 47.50±3.27 | 44.68 ±1.68 | 75.51 ±0.54 |
| HarmonyGNNs +Prob | 48.91±3.86 | 45.49±2.13 | 75.86±0.47 |

sure that the data split is consistent across all models. However, for those models among them that do not provide dataset-specific hyperparameters, such as MUSE, we conducted our own fine-tuning. For

other baselines, we derive the results from their original papers or baseline papers (Yuan et al., 2023; Xiao et al., 2024; Wan et al., 2024). For hyperparameter settings, see Appendix P.

## 4.1 LINEAR PROBING RESULTS OF OUR HARMONYGNNS

We present the performance comparisons of our HarmonyGNNs with state-of-the-art baseline methods across benchmarks in Table 1 and Table 2. The following observations can be made:

**Our HarmonyGNNs achieves significant improvement on heterophilic graph datasets, while retaining overall on-par performance on homophilic graph datasets.** *On heterophilic graph datasets*, compared to previous state-of-the-art graph SSL methods, our method outperforms all baselines—for example, by 7.12% on the Texas dataset, by 9.6% on the Roman-empire dataset, and by 1.27% on Actor. Similar observations hold when compared with previous SL methods; see Appendix C for a detailed analysis. *On the four homophilic graph datasets*, our HarmonyGNNs obtains better performance on Cora and Arxiv, on-par performance on CiteSeer and PubMed (with negligible performance drops that are within the standard deviations). The overall strong performance shows that our HarmonyGNNs is effective for both types of graphs.

**The two node-difficulty driven masking strategies in our HarmonyGNNs perform similarly.** The Bernoulli sampling based approach (i.e., HarmonyGNNs +Prob) is slightly better, thanks to its balance between exploration and exploitation. As we shall show in ablation studies (see Table 5), our proposed node-difficulty driven mask strategies are significantly better than the purely random masking strategy.

## 4.2 $k$-MEAN CLUSTERING RESULTS OF OUR HARMONYGNNS

From the clustering results in the Appendix D, our HarmonyGNNs achieves significantly better performance than all baselines, including the state-of-the-art model MUSE, by a large margin on the Texas and Cornell datasets, **with improvements of 11.26% and 12.51%**, respectively. Moreover, HarmonyGNNs slightly outperforms MUSE on Actor due to the complex mixed structural patterns, as introduced in Appendix N. It also attains comparable performance on Citeseer. These findings are consistent with those observed in linear probing based node classification tasks. Overall, our results demonstrate that HarmonyGNNs can generate high-quality embeddings regardless of the downstream tasks and effectively handle both heterophilic and homophilic patterns, highlighting its strong generalization capability in graph representation learning.

## 4.3 COMPUTE AND MEMORY COMPARISONS

To verify the efficiency of our proposed approach, we conducted an empirical analysis comparing our method to two major state-of-the-art SSL baselines: GREET and MUSE. As shown in Table 3, we measured

Table 3: Compute and Memory Comparisons

| Datasets | GPU MEMORY(MB) | | | EPOCH TIME(S/EPOCH) | | | TOTAL TIME(S) | | |
|---|---|---|---|---|---|---|---|---|---|
| | MUSE | GREET | OURS | MUSE | GREET | OURS | MUSE | GREET | OURS |
| Actor | 8786 | **4316** | 8608 | 0.43 | 0.56 | **0.23** | 53.64 | 64.32 | **28.98** |
| Roman | 34791 | 36425 | **29886** | 2.47 | 2.83 | **2.13** | 301.87 | 378.34 | **280.66** |
| Arxiv | 38791 | 35096 | **33486** | 5.31 | 6.71 | **4.03** | 627.32 | 738.86 | **476.16** |

memory usage, training time per epoch and total training time until convergence on three large scale datasets, Actor, Roman-Empire and Ogbn-Arxiv, that exhibit a complex mixture of patterns and require substantial computational resources. We utilized the optimal hyperparameters for each respective model. The results show that our HarmonyGNNs 's memory usage is on par with GREET (with only a slight increase for Actor at 4 GB) and remains lower than MUSE. Regarding running time, our HarmonyGNNs requires much less running time of the other two SOTA models while achieving much better performance, as shown in Table 1. This efficiency improvement is attributed to the fact that both GREET and MUSE employ an alternating training strategy for contrastive learning, which clearly highlights the advantages of our HarmonyGNNs .

Regarding the total training time until model convergence in the last column, our model is significantly faster than two baselines. By model convergence time, it means the time at which the best model is selected (out of the total number epochs that is the same for all models). This rapid convergence is attributable to the consistency in the latent space during reconstruction and end-to-end training—advantages that the baselines do not achieve.

Table 4: Results on Ablating Three Components.

| Methods | Cornell | Texas | Wisconsin | Actor | Roman | Cora | Citeseer | Pubmed | Arxiv |
|---|---|---|---|---|---|---|---|---|---|
| HarmonyGNNs (Full) | **85.68**±2.11 | **92.45**±3.78 | **93.13**±3.42 | **38.15**±0.71 | **75.86**±0.47 | **84.70**±0.56 | **73.36**±0.33 | **83.42**±0.26 | **71.56**±0.28 |
| w/o DynMsk | 84.26±2.15 | 90.16±3.51 | 90.08±3.36 | 36.98±0.87 | 74.01±0.50 | 84.10±0.85 | 72.90±0.53 | 81.98±0.63 | 71.00±0.56 |
| w/o T-S & DynMsk | 81.78±3.66 | 85.59±4.19 | 88.56±3.56 | 35.86±0.87 | 72.87±1.78 | 83.10±0.78 | 71.68±0.60 | 80.08±0.66 | 70.02±0.50 |
| w/o T-S & DynMsk & Attn | 79.86±3.82 | 82.46±5.05 | 86.98±3.60 | 34.11±0.92 | 70.12±1.89 | 78.36±0.80 | 69.60±0.56 | 78.05±0.60 | 68.65±0.58 |

## 4.4 ABLATION STUDIES

**Ablating Three Components**  Our HarmonyGNNs has three key components: a teacher-student predictive architecture (referred to *T-S*), node-difficulty driven dynamic masking strategies (referred to *DynMsk*), and encoding self-attention (referred to *Attn*). To evaluate the contribution of each individual component, we conduct an ablation study by progressively removing one component at a time. The results are shown in Table 4 and in Appendix E, we can observe,

- *DynMsk* can lead to performance decreases by up to 3.05% across the datasets when removed, which shows the effectiveness of the proposed node-difficulty driven masking strategies against purely random masking.
- *T-S* predictive architecture also plays a significant role, as performance drops considerably (1% - 4.57%) when we directly reconstruct the features in the raw input space using latent space features, as done in the encoder-decoder models, leading to a learning space mismatch. This observation is consistent with the theorem proposed in Sec. 3.4.
- Substituting *Attn* with a simple MLP also leads to performance drops noticeably. This indicates that attention fusion can also help adaptively assign weights to different components, allowing the model to effectively handle various patterns in graphs.

**Exploitation Ratio** $r$  We also evaluate performance under different exploitation ratios across datasets (Table 5) using the probabilistic masking scheme (Eqn. 4). Although the optimal masking ratio varies by dataset, dynamic masking consistently outperforms pure random masking ($r=0$), underscoring the need for our proposed dynamic masking and its integration with random masking.

Table 5: The effects of $r$ (Eqn. 4)

| Ratio $r$ | Cornell | Actor | Roman |
|---|---|---|---|
| 1 | 84.56±2.67 | 37.13±0.55 | 74.66±0.68 |
| 0.8 | **85.68**±2.11 | 37.93±0.61 | 75.34±0.45 |
| 0.5 | 85.34±2.75 | **38.15**±0.71 | **75.86**±0.47 |
| 0.3 | 84.40±2.60 | 37.70±0.78 | 74.88±0.48 |
| 0 | 84.26±2.15 | 36.98±0.87 | 74.01±0.50 |

**More Studies and Analysis**  More ablation studies covering WGCN impacts (App. K), the overall masking ratio (App. L), token-selection strategies (App. M) are provided in the Appendices.

## 5 CONCLUSION

In this paper we have presented HarmonyGNNs , a self-supervised framework designed to harmonize heterophily and homophily in GNNs. Through our joint structural node encoding, which integrates linear and non-linear feature transformations with K-hop structural embeddings, HarmonyGNNs adapts effectively to both homophilic and heterophilic graphs. Moreover, our teacher-student predictive paradigm, coupled with dynamic node-difficulty-based masking, further enhances robustness by providing progressively more challenging training signals. Comprehensive theoretical analysis and empirical results across benchmark datasets demonstrate that HarmonyGNNs consistently achieves state-of-the-art performance under heterophilic conditions using both linear probing and $k$-mean clustering evaluation protocols, while matching top methods on homophilic datasets. These findings underscore HarmonyGNNs 's capability to address the key challenges of capturing mixed structural properties, achieving superior performance without sacrificing efficiency.

### ACKNOWLEDGMENTS

This research was partly supported by ARO Grant W911NF1810295, NSF IIS-1909644, ARO Grant W911NF2210010, NSF CMMI-2024688, NSF IUSE-2013451 and NC State Goodnight Early Career Award. The views and conclusions contained herein are those of the authors and should not be interpreted as necessarily representing the official policies or endorsements, either expressed or implied, of ARO, NSF or the U.S. Government. The U.S. Government is authorized to reproduce and distribute reprints for Governmental purposes not withstanding any copyright annotation thereon.

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

# A RELATED WORK

## A.1 LEARNING ON HETEROPHILIC GRAPHS

Heterophilic graphs are prevalent in various domains, such as online transaction networks (Pandit et al., 2007), dating networks (Altenburger & Ugander, 2018), and molecular networks (Zhu et al., 2020a). Recently, significant efforts have been made to design novel GNNs that effectively capture information in heterophilic settings, where connected nodes possess dissimilar features and belong to different classes.

Some studies propose capturing information from long-range neighbors from various distance (Li et al., 2022; Liu et al., 2021; Abu-El-Haija et al., 2019; Pei et al., 2020; Suresh et al., 2021). For example, MixHop (Abu-El-Haija et al., 2019) concatenates information from multi-hop neighbors at each GNN layer. Geom-GCN (Pei et al., 2020) identifies potential neighbors in a continuous latent space. WRGAT (Suresh et al., 2021) captures information from distant nodes by defining the type and weight of edges across the entire graph to reconstruct a computation graph.

Other approaches focus on modifying traditional GNN architectures to achieve adaptive message passing from the neighborhood (Chen et al., 2020; Chien et al., 2020; Yan et al., 2021; Zhu et al., 2020a). For instance, GPR-GNN (Chien et al., 2020) incorporates learnable weights into the representations of each layer using the Generalized PageRank (GPR) technique, while H2GCN (Zhu et al., 2020a) removes self-loop connections and employs a non-mixing operation in the GNN layer to emphasize the features of the ego node.

Additionally, some papers approach the problem from spectral graph theory (Luan et al., 2021; Bo et al., 2021), claiming that high-pass filters can be beneficial in heterophilic graphs by sharpening the node features between neighbors and preserving high-frequency graph signals.

However, these methods still heavily rely on labeled data, which is impractical for real-world datasets due to the significant manual effort required and the necessity of ensuring label quality. Furthermore, they are limited in their ability to effectively learn from the data itself without extensive supervision.

## A.2 GRAPH REPRESENTATION LEARNING VIA SSL

As discussed in Section 1. traditional supervised learning on graphs suffers from performance degradation when labeled data is scarce. However, collecting and annotating manual labels in large-scale datasets (e.g., citation and social networks) is prohibitively expensive, or requires substantial domain expertise (e.g., chemistry and medicine). Additionally, these models are vulnerable to label-related noise, further undermining the robustness of graph semi-supervised learning. Self-supervised learning (SSL) has achieved widespread adoption in the fields of natural language processing (NLP) (Devlin et al., 2019) and computer vision (CV) (He et al., 2022; Assran et al., 2023). Unlike traditional supervised learning, which relies heavily on large amounts of labeled data, SSL leverages unlabeled data by creating proxy/pretext tasks that exploit intrinsic structures of raw data themselves as labels (such as the next word/token prediction, and masked word/image modeling). This approach not only addresses the dependency on the quantity of labeled data but also efficiently utilizes the inherent patterns and relationships within the data, enabling the development of richer representations without need for explicit annotations. Furthermore, they can also encourage the model to learn more robust representations, thereby reducing its sensitivity to noise and/or labeling bias. Building on these advantages, they have shown remarkable promise in various graph representation learning applications.

Because of the advantages mentioned above, self-supervised learning has also attracted significant attention in the field of graph representation learning. Graph SSL approaches are generally divided into two primary categories: graph contrastive learning and graph generative learning. (1) Contrastive Losses in Contrastive learning : The model is encouraged to bring representations of similar nodes (or augmented views) closer while pushing apart those of dissimilar nodes; (2) Feature/edge Reconstruction in generative learning: Given a masked input, the model is trained to reconstruct the original node features /predict the existence or weight of edges. However, both approaches become problematic under certain circumstances. Contrastive learning's success hinges on relatively complex training strategies, including the careful design of negative samples and a strong reliance on high-quality data augmentation (Grill et al., 2020). However, these requirements are often challenging to meet

in graph settings (Hou et al., 2022), which limits the broader application of contrastive learning in this domain. They can also suffer from representation collapse, where the network converges to a state where all outputs become similar, rendering the learned features uninformative. On the other hand, generative learning methods aim to reconstruct graph data but often face challenges due to reconstruction space mismatch. This arises because the decoder demands intricate design choices and frequently struggles to fully recover the original feature space (Hou et al., 2022; 2023). The decoder can also potentially inflate the model's parameter count and GPU memory footprint during training. Moreover, these methods are also prone to well-known issues such as training instability, overfitting, and mode collapse.

### A.2.1 GRAPH GENERATIVE LEARNING

Generation-based methods reconstruct graph data by focusing on either the features and the structure of the graph or both. Classic generation-based approaches include GAE (Kipf & Welling, 2016b), VGAE (Kipf & Welling, 2016b), and MGAE (Wang et al., 2017), which primarily aim to reconstruct the structural information of the graph, as well as S2GAE (Tan et al., 2023). In contrast, GraphMAE (Hou et al., 2022) and GraphMAE2 (Hou et al., 2023) utilize masked feature reconstruction as their primary objective, incorporating auxiliary designs to achieve performance that is comparable to or better than contrastive methods.

In the context of generative learning on heterophilic graphs, DSSL (Xiao et al., 2022) operates under the assumption of a graph generation process, decoupling diverse patterns to effectively capture high-order information. Similarly, NWR-GAE (Tang et al., 2022) jointly predicts the node degree and the distribution of neighbor features. However, despite these innovative approaches, their performance on node classification benchmarks is often unsatisfactory (Hou et al., 2022).

### A.2.2 GRAPH CONTRASTIVE LEARNING

Contrast-based methods generate representations from multiple views of a graph and aim to maximize their agreement, demonstrating effective practices in recent research. For example, DGI (Veličković et al., 2018) and InfoGraph (Sun et al., 2020) utilize node-graph mutual information maximization to capture both local and global information. MVGRL (Hassani & Khasahmadi, 2020) leverages graph diffusion to create an additional view of the graph and contrasts node-graph representations across these distinct views. GCC (Qiu et al., 2020) employs subgraph-based instance discrimination and adopts the InfoNCE loss as its pre-training objective. GRACE (Zhu et al., 2020b) and GraphCL (You et al., 2020) learn node or graph representations by maximizing the agreement between different augmentations while treating other nodes or graphs as negative instances. BGRL (Thakoor et al., 2022) contrasts two augmented versions using inter-view representations without relying on negative samples. Additionally, CCA-SSG (Zhang et al., 2021) adopts a feature-level objective for graph SSL, aiming to reduce the correlation between different views. These contrast-based approaches effectively harness the structural and feature information inherent in graph data, contributing to the advancement of self-supervised learning on graphs.

However, most of these methods are based on the homophily assumption. Recent works have demonstrated that SSL can also benefit heterophilic graphs. For instance, HGRL (Chen et al., 2022) enhances node representations on heterophilic graphs by reconstructing similarity matrices to generate two types of feature augmentations based on topology and features. GraphACL (Xiao et al., 2024) predicts the original neighborhood signal of each node using a predictor. MUSE (Yuan et al., 2023) constructs contrastive views by perturbing both the features and the graph topology, and it learns a graph-structure-based combiner. GREET (Liu et al., 2022) employs an edge discriminator to separate the graph into homophilic and heterophilic components, then applies low-pass and high-pass filters accordingly. However, these methods rely on the meticulous design of negative samples to provide effective contrastive signals. Moreover, although some approaches such as GREET and MUSE achieve impressive results, they require alternative training. This significantly increases computational overhead and may lead to suboptimal performance.

Note that, the fundamental goal of contrastive learning is to shape an embedding space in which similar (positive) samples are pulled together while dissimilar (negative) samples are pushed apart.

### A.2.3 SELF SUPERVISED LEARNING IN HARMONYGNNS :

Recently, the JEPA paradigm has shown remarkable effectiveness in self-supervised learning across diverse domains Assran et al. (2023). Motivated by its predictive objective principle, we design HarmonyGNNs as a graph-tailored framework that effectively tackles the challenges discussed above:

- No Negative Sampling. Our method requires no negative samples or positive–negative pair construction. This is a unique advantage of our model, as highlighted in Sec. 1. The student network's objective is to predict the teacher's output representations for all nodes, rather than to contrast pairs. We explicitly mention that we eliminate negative sampling, a core component of contrastive learning.
- No Contrastive Loss. We do not use contrastive loss functions (e.g., InfoNCE or NT-Xent). Equ. 2 defines a predictive loss in an aligned latent space, NOT a contrastive loss. We predict teacher network outputs for ALL nodes (both masked and unmasked), which is completely different from contrastive learning's paradigm of pulling positive pairs together and pushing negative pairs apart. We also provide a comprehensive theoretical analysis of this predictive architecture.
- Adaptive Node Masking. Our node masking is not mere data augmentation or random dropout. We introduce learnable parameters for masked nodes and adaptively select which nodes to mask based on prediction difficulty. This creates a more challenging, informative training task compared to uniform random node dropping.
- Teacher–Student All-Node Predictive Architecture. Only the student receives a masked view; the teacher always observes the full graph. This setup constitutes an information-completion task, not a dual-random-view contrastive training.

## B CHALLENGES OF HETEROPHILY AND HOMOPHILY FOR GRAPH REPRESENTATION LEARNING

In this section, we provide a preliminary analysis of the challenges involved in graph representation learning when handling a mixture of both heterophily and homophily patterns (see Tables 6 and 7 in the Appendix C). We examine current methods, including both semi-supervised learning (SL) and self-supervised learning (SSL) approaches.

- *SL methods:* GCN and GAT that focus on low-pass graph signals work well on the homophilic graph datasets, but suffer from significant performance drop on heterophilic graph datasets. WRGAT and H2GCN address these issues of GCN and GAT, leading to significant performance boost on heterophilic graph datasets, while retaining similar performance on homophilic graph datasets. To understand what the critical part is for performance improvement on the heterophilic graphs, and to test if high-pass signals indeed play a significant role for them, we test a vanilla MLP which totally ignores the topology of graphs (see Table 6), and simply uses the raw input node features. We can see the simple MLP works reasonably well on heterophilic datasets in comparisons with WRGAT and H2GCN, which supports our earlier statement that traditional message passing produces smoothing operations on the graph, highly relying on the homophily assumption, and highlights that the raw node features play a critical role in GNN learning on heterophilic graphs, whereas neighbor information is essential for learning on homophilic graphs. Overall, these observations motivate our joint structural node encoding (Eqn. 6). Meanwhile, MLP suffers from drastic performance drop on homophilic graph datasets, as expected.

- *Previous State-of-the-art SSL methods.* Those methods (DGI, GMI, MVGRL, BGRL, GRACE and GraphMAE) that are designed for homophilic graphs achieve significant progress in terms of bridging the SSL performance with the SL counterparts on homophilic graphs, but they inherit the drawbacks as GCN and GAT on heterophilic graphs. More recently, methods such as MUSE, GREET and S3GCL make promising improvement, but they do not show significant progress against the MLP SL baseline on heterophilic graphs, especially on Actor, which exhibits complex mixed patterns. Our HarmonyGNNs makes a step forward by significantly improving performance on heterophilic graphs, showing the great potential of graph SSL (see Table 1 and Table 2).

# C  Performance Comparisons with Semi-Supervised Learning Baselines

Similar as Table 1 and Table 2 in Section 4, we present the performance comparison with several prominent semi-supervised learning baselines in Tables 6 and Table 7, using the same datasets. The experimental settings—including data splits and labeling ratios for Cora, Citeseer, and Pubmed—are kept consistent across all experiments. For results of baselines, we use the results reported in (Platonov et al., 2023; Yuan et al., 2023). For evaluation, we still follow the linear-probing protocol: we freeze each model, generate embeddings, and train a downstream linear classifier for downstream node classification. We primarily compare against two groups of semi-supervised baselines:

- *Traditional supervised learning (SL) methods*: GCN (Kipf & Welling, 2016a), GAT (Veličković et al., 2017) and a simple MLP;

- *Supervised methods specifically designed for heterophilic graphs*: WRGAT (Suresh et al., 2021), H2GCN (Zhu et al., 2020a), GPR-GNN (Chien et al., 2020) and FAGCN (Bo et al., 2021).

Table 6: Results of node classification (in percent $\pm$ standard deviation across ten splits). The best and the runner-up results are highlighted in red and blue respectively in terms of the mean accuracy.

| | Methods | Heterophilic | | | | Homophilic | | | |
|---|---|---|---|---|---|---|---|---|---|
| | | Cornell | Texas | Wisconsin | Actor | Cora | CiteSeer | PubMed | Arxiv |
| SL | GCN (Kipf & Welling, 2016a) | 57.03±3.30 | 60.00±4.80 | 56.47±6.55 | 30.83±0.77 | 81.50±0.30 | 70.30±0.27 | 79.00±0.05 | 71.74±0.27 |
| | GAT (Veličković et al., 2017) | 59.46±3.63 | 61.62±3.78 | 54.71±6.87 | 28.06±1.48 | 83.02±0.19 | 72.51±0.22 | 79.87±0.03 | 71.92±0.17 |
| | MLP | 81.08±7.93 | 81.62±5.51 | 84.31±3.40 | 35.66±0.94 | 56.11±0.34 | 56.91±0.42 | 71.35±0.05 | 55.50±0.23 |
| | [†] WRGAT (Suresh et al., 2021) | 81.62±3.90 | 83.62±5.50 | 86.98±3.78 | 36.53±0.77 | 81.97±1.50 | 70.85±1.98 | 80.86±0.55 | — |
| | [†] H2GCN (Zhu et al., 2020a) | 82.16±4.80 | 84.86±6.77 | 86.67±4.69 | 35.86±1.03 | 81.76±1.55 | 70.53±2.01 | 80.26±0.56 | — |
| SSL-Ours | HarmonyGNNs +Diffi | 85.41±1.79 | 93.24±2.77 | 92.74±2.91 | 37.93±0.56 | 84.70±0.56 | 73.36±0.33 | 83.42±0.26 | 71.56±0.28 |
| | HarmonyGNNs +Prob | 85.68±2.11 | 92.45±3.78 | 93.13±3.42 | 38.15±0.71 | 84.82±0.23 | 73.12±0.28 | 83.25±0.16 | 71.97±0.12 |

[†] Neither WRGAT nor H2GCN have available hyperparameter configurations specifically tuned for the OGBN-Arxiv dataset in their original paper or baseline papers.

Table 7: Results of node classification (in percent $\pm$ standard deviation across ten splits). The best and the runner-up results are highlighted in red and blue respectively in terms of the mean accuracy.

| | Methods | Chameleon(filtered) | Squirrel(filtered) | Roman-Empire |
|---|---|---|---|---|
| SL | GCN (Kipf & Welling, 2016a) | 40.89±4.12 | 39.47±1.47 | 73.69±0.74 |
| | GPR-GNN (Chien et al., 2020) | 39.93±3.30 | 38.95±1.99 | 64.85±0.27 |
| | FAGCN (Bo et al., 2021) | 41.90±2.72 | 41.08±2.27 | 65.22±0.56 |
| | H2GCN (Zhu et al., 2020a) | 26.75±3.64 | 35.10±1.15 | 60.11±0.52 |
| SSL-Ours | HarmonyGNNs +Diffi | 47.50±3.27 | 44.68±1.68 | 75.51±0.54 |
| | HarmonyGNNs +Prob | 48.91±3.86 | 45.49±2.13 | 75.86±0.47 |

From the results, we draw the same conclusion as in our comparison with SSL baselines in the main text: our HarmonyGNNs consistently outperforms all SL baselines on heterophilic datasets—including both classical GNNs and models specifically designed for heterophily—while achieving comparable performance on homophilic datasets. For example, HarmonyGNNs surpasses the strongest baselines by 8.38% on Texas, 6.15% on Wisconsin 4.41% on filtered squirrel and by 7.01% on filtered Chameleon, demonstrating its ability to learn complex mixed patterns in graphs. Moreover, when comparing the two masking strategies, probabilistic masking consistently outperforms difficulty-based masking. This suggests that applying a base masking probability to all nodes—rather than focusing solely on difficult ones during training—more effectively balances exploration and exploitation. This observation is consistent with the conclusion drawn in the main text.

# D  Performance Comparison for Node Clustering

In this section, we present a performance comparison for node clustering. We compare our model with four groups of baseline methods:

Table 8: Clustering results (ACC in percent $\pm$ standard deviation). The best and runner-up results are highlighted with red and blue, respectively.

| Methods | Texas | Actor | Cornell | CiteSeer |
|---|---|---|---|---|
| | ACC | ACC | ACC | ACC |
| AE (Hinton & Salakhutdinov, 2006) | 50.49±0.01 | 24.19±0.11 | 52.19±0.01 | 58.79±0.19 |
| node2vec (Grover & Leskovec, 2016) | 48.80±1.93 | 25.02±0.04 | 50.98±0.01 | 20.76±0.27 |
| struc2vec (Ribeiro et al., 2017) | 49.73±0.01 | 22.49±0.34 | 32.68±0.01 | 21.22±0.45 |
| LINE (Tang et al., 2015) | 49.40±2.08 | 22.70±0.08 | 34.10±0.77 | 28.42±0.88 |
| GAE (Kipf & Welling, 2016b) | 42.02±1.22 | 23.45±0.04 | 43.72±1.25 | 48.37±0.37 |
| VGAE (Kipf & Welling, 2016b) | 50.27±1.87 | 23.30±0.22 | 43.39±0.99 | 55.67±0.13 |
| GraphSAGE (Hamilton et al., 2017) | 56.83±0.56 | 23.08±0.29 | 44.70±2.00 | 49.28±1.18 |
| SDCN (Bo et al., 2020) | 44.04±0.56 | 23.67±0.29 | 36.94±2.00 | 59.86±1.18 |
| MVGRL (Hassani & Khasahmadi, 2020) | 62.79±2.33 | 28.58±1.03 | 43.77±3.03 | 45.67±9.08 |
| GRACE (Zhu et al., 2020b) | 56.99±2.23 | 25.87±0.45 | 43.55±4.60 | 54.66±5.41 |
| BGRL (Thakoor et al., 2021) | 58.68±1.80 | 28.20±0.27 | 55.08±1.68 | 64.27±1.68 |
| DSSL (Xiao et al., 2022) | 57.43±3.51 | 26.15±0.46 | 44.70±2.44 | 54.32±3.69 |
| HGRL (Chen et al., 2022) | 61.97±3.10 | 29.79±1.11 | 60.56±3.72 | 61.14±1.49 |
| [†] MUSE (Yuan et al., 2023) | 65.79±4.36 | 31.05±0.72 | 62.35±2.38 | 66.03±2.33 |
| HarmonyGNNs +Diffi | 76.50±1.50 | 31.22±0.76 | 73.22±3.45 | 65.80±2.32 |
| HarmonyGNNs +Prob | 77.05±2.66 | 32.10±1.51 | 74.86±2.09 | 66.56±3.56 |

[†] MUSE doesn't provide any hyperparameters for node clustering.

- *Traditional Unsupervised Clustering Methods:* AE (Hinton & Salakhutdinov, 2006), node2vec (Grover & Leskovec, 2016), struc2vec (Ribeiro et al., 2017), and LINE (Tang et al., 2015).

- *Attributed Graph Clustering Methods:* GAE (VGAE) (Kipf & Welling, 2016b), GraphSAGE (Hamilton et al., 2017), and SDCN (Bo et al., 2020).

- *Self Supervised Methods for Homophilic Graphs:* MVGRL (Hassani & Khasahmadi, 2020), GRACE (Zhu et al., 2020b), and BGRL (Thakoor et al., 2021).

- *Self Supervised Methods for Heterophilic Graphs:* DSSL (Xiao et al., 2022), HGRL (Chen et al., 2022), and MUSE (Yuan et al., 2023).

Following the same protocol as with other baselines, we freeze the model and use the generated embeddings for *k*-means clustering. We reproduce MUSE (Yuan et al., 2023), as it has been proven to be the state-of-the-art model for node clustering. However, the original paper does not provide any hyperparameters for node clustering on any dataset, we perform hyperparameter tuning ourselves. For the other baselines, we report the results from baseline papers (Chen et al., 2022; Yuan et al., 2023). The hyperparameters search space can be found in Appendix P. The results are shown in Table 8.

From the results, we can achieve the similar conclusions as node classification:

- Our HarmonyGNNs achieves significantly better performance than all baselines, including the state-of-the-art model MUSE, by a large margin on the Texas and Cornell datasets, with improvements of 11.26% and 12.51%, respectively. Moreover, HarmonyGNNs slightly outperforms MUSE on Actor due to the complex mixed structural patterns, as introduced in Appendix N. It also attains comparable performance on Citeseer. These findings are consistent with those observed in node classification tasks. Overall, our results demonstrate that HarmonyGNNs can generate high-quality embeddings regardless of the downstream tasks and effectively handle both heterophilic and homophilic patterns, highlighting its strong generalization capability in graph representation learning.

- Regarding the two masking strategies, probabilistic masking consistently outperforms difficulty masking. This finding aligns with our observations in node classification and can be attributed to a better balance between exploration and exploitation.

Table 9: Ablation study on heterophilic datasets. Accuracy (%) with mean $\pm$ std.

| Methods | Cornell | Texas | Wisconsin | Actor | Roman |
|---|---|---|---|---|---|
| HarmonyGNNs (Full) | **85.68**$\pm$2.11 | **92.45**$\pm$3.78 | **93.13**$\pm$3.42 | **38.15**$\pm$0.71 | **75.86**$\pm$0.47 |
| w/o DynMsk | 84.26$\pm$2.15 | 90.16$\pm$3.51 | 90.08$\pm$3.36 | 36.98$\pm$0.87 | 74.01$\pm$0.50 |
| w/o T-S | 82.09$\pm$2.85 | 87.96$\pm$3.87 | 89.02$\pm$3.12 | 36.08$\pm$1.02 | 73.11$\pm$1.12 |
| w/o Attn | 82.85$\pm$2.33 | 88.96$\pm$4.00 | 90.23$\pm$3.26 | 37.02$\pm$0.62 | 73.53$\pm$1.36 |
| w/o T-S & DynMsk | 81.78$\pm$3.66 | 85.59$\pm$4.19 | 88.56$\pm$3.56 | 35.86$\pm$0.87 | 72.87$\pm$1.78 |
| w/o T-S & DynMsk & Attn | 79.86$\pm$3.82 | 82.46$\pm$5.05 | 86.98$\pm$3.60 | 34.11$\pm$0.92 | 70.12$\pm$1.89 |

## E ABLATION STUDY ON PROPOSED TECHNIQUES

We perform an ablation to illustrate the interactions between our masking strategies and the other model components in Table 9.

Results show that dynamic masking and the teacher–student predictive architecture usually interact: the performance drop from removing both is not simply the sum of their individual effects, underscoring their interdependence. As noted in the Sec. 3.1, masking strategies are critical to SSL's success.

However, dynamic masking and attention usually operate orthogonally: dynamic masking informs SSL of complex, often unknown topological properties of graphs, while attention fuses multiple filters to capture complex structural patterns. The results in the table also align with our expectations.

## F PERFORMANCE COMPARISON WITH RECENT BASELINES

In this section, we present a performance comparison of node classification using recent and strong state-of-the-art SL baselines, namely PCNet (Li et al., 2024), AEROGNN (Lee et al., 2023), and $G^2$ (Rusch et al., 2022). We report the results as provided in their respective original papers. Because prior works evaluate on different datasets (e.g., $G^2$ only on heterophilic graphs, while PC-Conv adopts different splits on homophilic benchmarks), we restrict our comparisons to identical settings for fairness. Therefore, we report results for all methods on heterophilic datasets and include AeroGNN on three homophilic datasets, as it is the only method evaluated under the same experimental protocol as ours and presented in Table 1. As shown in Table 10 and 11, Our HarmonyGNNs achieves state-of-the-art performance on the Wisconsin, Texas, and Actor datasets, as well as on three homophilic benchmarks, while maintaining competitive results on Cornell. This indicates HarmonyGNNs 's ability to handle complex mixed patterns in graphs.

Table 10: Results of node classification (in percent $\pm$ standard deviation across ten splits). The best and the runner-up results are highlighted in red and blue respectively in terms of the mean accuracy.

| Methods | Cornell | Texas | Wisconsin | Actor |
|---|---|---|---|---|
| | ACC | ACC | ACC | ACC |
| PCNet (Li et al., 2024) | 82.16$\pm$ 2.70 | 88.11$\pm$2.17 | 88.63$\pm$ 2.75 | 37.80$\pm$ 0.64 |
| AEROGNN (Lee et al., 2023) | 81.24$\pm$6.80 | 84.35$\pm$5.20 | 84.80$\pm$3.30 | 36.57$\pm$1.10 |
| [†] $G^2$ (Rusch et al., 2022) | 86.22$\pm$4.90 | 87.57$\pm$3.86 | 87.84$\pm$3.49 | — |
| S3GCL (Wan et al., 2024) | 81.27$\pm$3.67 | 86.12$\pm$3.91 | 84.56$\pm$2.71 | 36.88$\pm$0.34 |
| HarmonyGNNs +Diffi | 85.41$\pm$1.79 | 93.24$\pm$2.77 | 92.74$\pm$2.91 | 37.93$\pm$0.56 |
| HarmonyGNNs +Prob | 85.68$\pm$2.11 | 92.45$\pm$3.78 | 93.13$\pm$3.42 | 38.15$\pm$0.71 |

[†] $G^2$ has no reported performance on the Actor dataset.

Table 11: Results of node classification (in percent $\pm$ standard deviation across ten splits). The best and the runner-up results are highlighted in red and blue respectively in terms of the mean accuracy.

| Methods | Cora | Citeseer | Pubmed |
|---------|------|----------|--------|
| AEROGNN (Lee et al., 2023) | $83.90 \pm 0.50$ | $73.20 \pm 0.60$ | $80.59 \pm 0.50$ |
| HarmonyGNNs + Diff | $84.70 \pm 0.56$ | $73.36 \pm 0.33$ | $83.42 \pm 0.26$ |
| HarmonyGNNs + Prob | $84.82 \pm 0.23$ | $73.12 \pm 0.28$ | $83.25 \pm 0.16$ |

## G  PROOF OF GRADIENT-DIFFERENCE BOUNDS

**Theorem 2.** *Consider the optimization of encoder-decoder based graph SSL in Eqn. 8 and our proposed HarmonyGNNs in Eqn. 2 under the same encoder architecture and following assumptions/conditions:*

- ***Gradient Smoothness and Lipschitz Continuity** for the encoder, the decoder, E.g., the encoder $E(\cdot; \Theta_{enc})$ has gradient $\beta_E$-smoothness (i.e., each gradient from iteration $t$ to $t+1$ changes at most linearly with respect to parameter shifts in $\Theta_{enc}$ with a coefficient $\beta_E$) and is $L_E$-Lipschitz continuous with respect to its input and/or parameters (i.e., differences such as $||E(\cdot; \Theta_{enc}^{(t+1)}) - E(\cdot; \Theta_{enc}^{(t)})||$ can be bounded from the above as linear functions of $||\Theta_{enc}^{(t+1)} - \Theta_{enc}^{(t)}||$ with a coefficient $L_E$). Similarly, we have $(\beta_D, L_D)$ defined for the decoder.*

- ***Boundedness** from the above for gradients of the encoder, gradients of the decoder, and reconstruction errors of the combined encoder-decoder.*

  *So, $||\nabla E(\cdot; \Theta_{enc}^{(t)})|| \leq B_E$, $||\nabla D(E(\cdot; \Theta_{enc}^{(t)}); \Theta_{dec}^{(t)})|| \leq B_D$, and $||D(E(\bar{f}; \Theta_{enc}^{(t)}); \Theta_{dec}^{(t)}) - f|| \leq B_{Reconst}$.*

- ***Strong Convexity** for the encoder, the decoder, and the student (and the teacher) in their parameters.*

  *E.g., the encoder $E(\cdot; \Theta_{enc})$ is $\mu_E$-strongly convex in their parameters $\Theta_{enc}$, i.e., $\langle \nabla E(\bar{f}; \Theta_{enc}^{(t+1)}) - \nabla E(\bar{f}; \Theta_{enc}^{(t)}), \Theta_{enc}^{(t+1)} - \Theta_{enc}^{(t)} \rangle \geq \mu_E \cdot ||\Theta_{enc}^{(t+1)} - \Theta_{enc}^{(t)}||^2$. Similarly, we have $\mu_D$ defined for the decoder.*

- ***Approximation Error.** When only unmasked inputs are used, the composite functions, either the encoder-decoder or the teacher-student in our HarmonyGNNs, achieve an approximation error $\epsilon_{E-D}$ (or $\epsilon_{T-S}$).*

*Then, the following three results hold:*

- ***Linear Convergence Bounds Under Strong Convexity.** For our HarmonyGNNs,*

$$||\Phi^{(t+1)} - \Phi^*||^2 \leq (1 - \frac{\mu_E^2}{\beta_E^2}) \cdot ||\Phi^{(t)} - \Phi^*||^2 \tag{15}$$

  *For the encoder-decoder models,*

$$||\theta^{(t+1)} - \theta^*||^2 \leq \left(1 - \frac{\min(\mu_E^2, \mu_D^2)}{\max(\beta_E^2, \beta_D^2)}\right) ||\theta^{(t)} - \theta^*||^2 \tag{16}$$

  *from which we can see our HarmonyGNNs converges to the optimal solution $\Phi^*$ faster than the encoder-decoder counterpart to their optimal solutions $\Theta^*$ due to a smaller contraction factor $\left(1 - \frac{\mu_E^2}{\beta_E^2}\right) < \left(1 - \frac{\min(\mu_E^2, \mu_D^2)}{\max(\beta_E^2, \beta_D^2)}\right)$. This implies that HarmonyGNNs can achieve a faster convergence.*

- ***Proxy Task Loss Bounds** under a Lipschitz-dependent assumption between the masked graph signal and the raw graph signal, $||\bar{f} - f|| \leq \delta$. For our HarmonyGNNs,*

$$||S(\bar{f}; \Phi) - T(f; \Psi)|| \leq L_E \cdot \delta + \epsilon_{T-S}. \tag{17}$$

  *For the encoder-decoder models,*

$$||D(E(\bar{f}; \Phi_{enc}); \Theta_{dec}) - f|| \leq L_E \cdot L_D \cdot \delta + \epsilon_{E-D}. \tag{18}$$

*W.L.O.G., assume $\epsilon_{E-D} = \epsilon_{T-S}$, our HarmonyGNNs has a smaller error upper bound, indicating that our teacher–student model is closer to the optimal solution $\theta^*$ during training, which in turn implies that its parameter updates are more stable and its convergence speed is faster (as shown in the first result above).*

- **Gradient-Difference Bounds** *in Encoder-Decoder Models Showing Coupling Effects of Parameter Updating,*

$$\|\nabla\mathcal{L}_{\text{E}-D}(\Theta_{enc}^{(t+1)}) - \nabla\mathcal{L}_{E-D}(\Theta_{enc}^{(t)})\| \leq 2B_{Reconst}\Big(\beta_E B_D + B_E L_D L_E\Big)\|\Theta_{enc}^{(t+1)} - \Theta_{enc}^{(t)}\|$$
(19)

$$+ 2B_E B_{Reconst}\beta_D\|\Theta_{dec}^{(t+1)} - \Theta_{dec}^{(t)}\| + 4B_E B_D B_{Reconst},$$

$$\|\nabla\mathcal{L}_{\text{E}-D}(\Theta_{dec}^{(t+1)}) - \nabla\mathcal{L}_{E-D}(\Theta_{dec}^{(t)})\| \leq 2B_{Reconst}\,\beta_D\,L_E\,\|\Theta_{enc}^{(t+1)} - \Theta_{enc}^{(t)}\|$$
(20)

$$+ 2B_{Reconst}\beta_D\|\Theta_{dec}^{(t+1)} - \Theta_{dec}^{(t)}\| + 4B_D B_{Reconst},$$

*where the coupling effects in Encoder-Decoder models may lead to instability in learning.*

**In this section, we first provide the proof of the Gradient Difference Upper Bound:**

### G.1 ENCODER GRADIENT DIFFERENCE UPPER BOUND IN ENCODER-DECODER MODEL:

Consider the encoder-decoder model loss function

$$\mathcal{L}_{E-D}(\Theta) = \frac{1}{N}\|D\big(E(\bar{f};\Theta_{enc});\Theta_{dec}\big) - f\|_2^2$$
(21)

Assume the following:

1. **Encoder Smoothness:**
$$\|\nabla E(\cdot;\Theta_{enc}^{(t+1)}) - \nabla E(\cdot;\Theta_{enc}^{(t)})\| \leq \beta_E\,\|\Theta_{enc}^{(t+1)} - \Theta_{enc}^{(t)}\|.$$
(22)

2. **Decoder Gradient Smoothness:** For any fixed input (e.g. $f_{\theta_f}(\overline{x})$),
$$\Big\|\nabla D^{(t+1)}\big(E(\cdot;\Theta_{enc}^{(t+1)})\big)\big) - \nabla D\big(E(\cdot;\Theta_{enc}^{(t)});\Theta_{dec}^{(t)}\big)\Big\| \leq \beta_D\,\|\Theta_{dec}^{(t+1)} - \Theta_{dec}^{(t)}\|$$
(23)
$$+ L_D L_E\,\|\Theta_{enc}^{(t+1)} - \Theta_{enc}^{(t)}\|,$$

3. **Encoder Gradient Bound:**
$$\|\nabla E(\cdot;\Theta_{enc}^{(t)})\| \leq B_E.$$
(24)

4. **Decoder Gradient Bound:**
$$\Big\|\nabla D\big(E(\cdot;\Theta_{enc}^{(t)});\Theta_{dec}^{(t)}\big)\Big\| \leq B_D.$$
(25)

   We use the simplified notation in our proof:
$$\Big\|\nabla D^{(t)}\big(E(\cdot;\Theta_{enc}^{(t)})\big)\Big\| \leq B_D$$
(26)

5. **Reconstruction Error Bound:**
$$\Big\|D\big(E(\cdot;\Theta_{enc}^{(t)});\Theta_{dec}^{(t)} - f\Big\| \leq B_{Reconst}.$$
(27)

6. **Encoder Lipschitz (with respect to parameters):** There exists $L_E > 0$ such that
$$\|E(\cdot;\Theta_{enc}^{(t+1)}) - E(\cdot;\Theta_{enc}^{(t)})\| \leq L_E\,\|\Theta_{enc}^{(t+1)} - \Theta_{enc}^{(t)}\|.$$
(28)

Then, the gradient difference with respect to the encoder parameters between two consecutive iterations is bounded by

$$\Big\|\nabla\mathcal{L}_{\text{E}-\text{D}}(\Theta_{enc}^{(t+1)}) - \nabla\mathcal{L}_{E-D}(\Theta_{enc}^{(t)})\Big\| \leq C_1\,\|\Theta_{enc}^{(t+1)} - \Theta_{enc}^{(t)}\| + C_2\,\|\Theta_{dec}^{(t+1)} - \Theta_{dec}^{(t)}\| + C_3,$$
(29)

where

$$C_1 = 2B_{Reconst}\Big(\beta_E B_D + B_E L_D L_E\Big), \qquad C_2 = 2B_E\,\beta_D\,B_{Reconst}, \qquad C_3 = 4B_E B_D B_{Reconst}$$
(30)

*Proof.* We start with the expression for the gradient with respect to the encoder parameters at iteration $t$:

$$\nabla \mathcal{L}_{\text{E-D}}(\Theta_{enc}^{(t)}) = 2\left[D^{(t)}\big(E(\cdot;\Theta_{enc}^{(t)})\big) - f\right]\nabla E(\cdot;\Theta_{enc}^{(t)})\,\nabla D^{(t)}\big(E(\cdot;\Theta_{enc}^{(t)})\big). \tag{31}$$

Similarly, at iteration $t+1$,

$$\nabla \mathcal{L}_{\text{E-D}}(\Theta_{enc}^{(t+1)}) = 2\left[D^{(t+1)}\big(E(\cdot;\Theta_{enc}^{(t+1)})\big) - f\right]\nabla E(\cdot;\Theta_{enc}^{(t+1)})\,\nabla D^{(t+1)}\big(E(\cdot;\Theta_{enc}^{(t+1)})\big). \tag{32}$$

Define the difference:

$$\Delta_f = \left\|\nabla \mathcal{L}_{\text{E-D}}(\Theta_{enc}^{(t+1)}) - \nabla \mathcal{L}_{E-D}(\Theta_{enc}^{(t)})\right\|. \tag{33}$$

Thus,

$$\begin{aligned}
\Delta_f = \Big\| &2\,\nabla E(\cdot;\Theta_{enc}^{(t+1)})\,\nabla D^{(t+1)}\big(E(\cdot;\Theta_{enc}^{(t+1)})\big)\Big[D^{(t+1)}\big(E(\cdot;\Theta_{enc}^{(t+1)})\big) - f\Big] \\
&- 2\Big[D^{(t)}\big(E(\cdot;\Theta_{enc}^{(t)})\big) - f\Big]\nabla E(\cdot;\Theta_{enc}^{(t)})\,\nabla D^{(t)}\big(E(\cdot;\Theta_{enc}^{(t)})\big)\Big\|.
\end{aligned} \tag{34}$$

To handle this difference, we add and subtract the intermediate term

$$2\,\nabla E(\cdot;\Theta_{enc}^{(t)})\,\nabla D^{(t+1)}\big(E(\cdot;\Theta_{enc}^{(t+1)})\big)\Big[D^{(t+1)}\big(E(\cdot;\Theta_{enc}^{(t+1)})\big) - f\Big], \tag{35}$$

so that

$$\begin{aligned}
\Delta_f = \Big\| &2\Big[\nabla E(\cdot;\Theta_{enc}^{(t+1)}) - \nabla E(\cdot;\Theta_{enc}^{(t)})\Big]\nabla D^{(t+1)}\big(E(\cdot;\Theta_{enc}^{(t+1)})\big)\Big(D^{(t+1)}\big(E(\cdot;\Theta_{enc}^{(t+1)})\big) - f\Big) \\
&+ 2\,\nabla E(\cdot;\Theta_{enc}^{(t)})\Big\{\nabla D^{(t+1)}\big(E(\cdot;\Theta_{enc}^{(t+1)})\big) - \nabla D^{(t)}(E(\cdot;\Theta_{enc}^{(t)}))\Big\}\Big(D^{(t+1)}\big(E(\cdot;\Theta_{enc}^{(t+1)})\big) - f\Big) \\
&+ 2\,\nabla E(\cdot;\Theta_{enc}^{(t)})\,\nabla D^{(t)}(E(\cdot;\Theta_{enc}^{(t)}))\Big\{\Big(D^{(t+1)}\big(E(\cdot;\Theta_{enc}^{(t+1)})\big) - f\Big) - \Big(D^{(t)}\big(E(\cdot;\Theta_{enc}^{(t)})\big) - f\Big)\Big\}\Big\|.
\end{aligned} \tag{36}$$

Applying the triangle inequality yields:

$$\Delta_f \le T_1 + T_2 + T_3, \tag{37}$$

with

$$T_1 = 2\left\|\nabla E(\cdot;\Theta_{enc}^{(t+1)}) - \nabla E(\cdot;\Theta_{enc}^{(t)})\right\|\left\|\nabla D^{(t+1)}\big(E(\cdot;\Theta_{enc}^{(t+1)})\big)\right\|\left\|D^{(t+1)}\big(E(\cdot;\Theta_{enc}^{(t+1)})\big) - f\right\|, \tag{38}$$

and

$$T_2 = 2\left\|\nabla E(\cdot;\Theta_{enc}^{(t)})\right\|\left\|\nabla D^{(t+1)}\big(E(\cdot;\Theta_{enc}^{(t+1)})\big) - \nabla D^{(t)}\big(E(\cdot;\Theta_{enc}^{(t)})\big)\right\|\left\|D^{(t+1)}\big(E(\cdot;\Theta_{enc}^{(t+1)}) - f\right\|. \tag{39}$$

and

$$T_3 = 2\|\nabla E(\cdot;\Theta_{enc}^{(t)})\,\nabla D^{(t)}(E(\cdot;\Theta_{enc}^{(t)}))\Big\{\Big(D^{(t+1)}\big(E(\cdot;\Theta_{enc}^{(t+1)})\big) - f\Big) - \Big(D^{(t)}\big(E(\cdot;\Theta_{enc}^{(t)})\big) - f\Big)\Big\}\Big\|. \tag{40}$$

**Bounding $T_1$:** By the encoder smoothness assumption,

$$\|\nabla E(\cdot;\Theta_{enc}^{(t+1)}) - \nabla E(\cdot;\Theta_{enc}^{(t)})\| \le \beta_E\,\|\Theta_{enc}^{(t+1)} - \Theta_{enc}^{(t)}\|, \tag{41}$$

and by the decoder gradient bound,

$$\left\|\nabla D^{(t+1)}\big(E(\cdot;\Theta_{enc}^{(t+1)})\big)\right\| \le B_D, \tag{42}$$

and the reconstruction error bound,

$$\left\|D^{(t+1)}\big(E(\cdot;\Theta_{enc}^{(t+1)})\big) - f\right\| \le B_{Reconst}. \tag{43}$$

Thus,

$$T_1 \leq 2\,\beta_E\,B_D\,B_{Reconst}\,\|\Theta_{enc}^{(t+1)} - \Theta_{enc}^{(t)}\|. \tag{44}$$

**Bounding $T_2$:** We now decompose the term

$$\nabla D^{(t+1)}\big(E(\cdot;\Theta_{enc}^{(t+1)})\big) - \nabla D^{(t)}\big(E(\cdot;\Theta_{enc}^{(t)})\big). \tag{45}$$

By adding and subtracting the term $\nabla D^{(t+1)}((E(\cdot;\Theta_{enc}^{(t)})))$, we obtain:

$$\left\|\nabla D^{(t+1)}\big(E(\cdot;\Theta_{enc}^{(t+1)})\big) - \nabla D^{(t)}\big(E(\cdot;\Theta_{enc}^{(t)})\big)\right\|$$
$$\leq \left\|\nabla D^{(t+1)}\big(E(\cdot;\Theta_{enc}^{(t+1)})\big) - \nabla D^{(t+1)}((E(\cdot;\Theta_{enc}^{(t)})))\right\| \tag{46}$$
$$+ \left\|\nabla D^{(t+1)}((E(\cdot;\Theta_{enc}^{(t)}))) - \nabla D^{(t)}\big(E(\cdot;\Theta_{enc}^{(t)})\big)\right\|.$$

By the decoder's Lipschitz continuity with respect to its input, we have:

$$\left\|\nabla D^{(t+1)}\big(E(\cdot;\Theta_{enc}^{(t+1)})\big) - \nabla D^{(t+1)}((E(\cdot;\Theta_{enc}^{(t)})))\right\| \leq L_D\,\|E(\cdot;\Theta_{enc}^{(t+1)}) - E(\cdot;\Theta_{enc}^{(t)})\|, \tag{47}$$

and by the encoder Lipschitz condition,

$$\|E(\cdot;\Theta_{enc}^{(t+1)}) - E(\cdot;\Theta_{enc}^{(t)})\| \leq L_E\,\|\Theta_{enc}^{(t+1)} - \Theta_{enc}^{(t)}\|. \tag{48}$$

Thus, the first term is bounded by:

$$L_D L_E\,\|\Theta_{enc}^{(t+1)} - \Theta_{enc}^{(t)}\|. \tag{49}$$

For the second term, the decoder gradient smoothness gives:

$$\left\|\nabla D^{(t+1)}((E(\cdot;\Theta_{enc}^{(t)}))) - \nabla D^{(t)}((E(\cdot;\Theta_{enc}^{(t)})))\right\| \leq \beta_D\,\|\Theta_{dec}^{(t+1)} - \Theta_{dec}^{(t)}\|. \tag{50}$$

Thus,

$$\left\|\nabla D^{(t+1)}\big(E(\cdot;\Theta_{enc}^{(t+1)})\big) - \nabla D^{(t)}\big(E(\cdot;\Theta_{enc}^{(t)})\big)\right\| \leq L_D L_E\,\|\Theta_{enc}^{(t+1)} - \Theta_{enc}^{(t)}\| + \beta_D\,\|\Theta_{dec}^{(t+1)} - \Theta_{dec}^{(t)}\|. \tag{51}$$

Now, using the encoder gradient bound, $\|\nabla E(\cdot;\Theta_{enc}^{(t)})\| \leq B_E$, and the reconstruction error bound $\|D^{(t+1)}\big(E(\cdot;\Theta_{enc}^{(t+1)})\big) - f\| \leq B_{Reconst}$, we have:

$$T_2 \leq 2\,B_E\,B_{Reconst}\left(L_D L_E\,\|\Theta_{enc}^{(t+1)} - \Theta_{enc}^{(t)}\| + \beta_D\,\|\Theta_{dec}^{(t+1)} - \Theta_{dec}^{(t)}\|\right). \tag{52}$$

**Bounding $T_3$:**

$$T_3 = 2\|\nabla E(\cdot;\Theta_{enc}^{(t)})\,\nabla D^{(t)}(E(\cdot;\Theta_{enc}^{(t)}))\Big\{\Big(D^{(t+1)}\big(E(\cdot;\Theta_{enc}^{(t+1)})\big) - f\Big) - \Big(D^{(t)}\big(E(\cdot;\Theta_{enc}^{(t)})\big) - f\Big)\Big\}\| \tag{53}$$

$$\leq 2\|\nabla E(\cdot;\Theta_{enc}^{(t)})\| \cdot \|\nabla D^{(t)}(E(\cdot;\Theta_{enc}^{(t)}))\| \cdot \|\Big(D^{(t+1)}\big(E(\cdot;\Theta_{enc}^{(t+1)})\big) - f\Big) - \Big(D^{(t)}\big(E(\cdot;\Theta_{enc}^{(t)})\big) - f\Big)\| \tag{54}$$

$$\leq 2B_E \cdot B_D \cdot 2B_{Reconst} \tag{55}$$
$$= 4B_E B_D B_{Reconst} \tag{56}$$

**Combining $T_1$, $T_2$ and $T_3$:**

$$\Delta_f \leq T_1 + T_2 + T_3$$
$$\leq 2\beta_E B_D B_{Reconst}\,\|\Theta_{enc}^{(t+1)} - \Theta_{enc}^{(t)}\| + 2B_E B_{Reconst}L_D L_E\,\|\Theta_{enc}^{(t+1)} - \Theta_{enc}^{(t)}\|$$
$$+ 2B_E B_{Reconst}\beta_D\,\|\Theta_{dec}^{(t+1)} - \Theta_{dec}^{(t)}\| + 4B_E B_D B_{Reconst}$$
$$= \Big[2B_{Reconst}\big(\beta_E B_D + B_E L_D L_E\big)\Big]\,\|\Theta_{enc}^{(t+1)} - \Theta_{enc}^{(t)}\| + 2B_E B_{Reconst}\beta_D\,\|\Theta_{dec}^{(t+1)} - \Theta_{dec}^{(t)}\|$$
$$+ 4B_E B_D B_{Reconst}.$$

Define

$$C_1 = 2B_{Reconst}\Big(\beta_E B_D + B_E L_D L_E\Big) \quad \text{and} \quad C_2 = 2B_E B_{Reconst}\beta_D \quad \text{and} \quad C_3 = 4B_E B_D B_{Reconst}. \tag{57}$$

Then, the final bound is:

$$\Big\|\nabla\mathcal{L}_{\text{E-D}}(\Theta_{enc}^{(t+1)}) - \nabla\mathcal{L}_{E-D}(\Theta_{enc}^{(t)})\Big\| \leq C_1 \|\Theta_{enc}^{(t+1)} - \Theta_{enc}^{(t)}\| + C_2 \|\Theta_{dec}^{(t+1)} - \Theta_{dec}^{(t)}\| + 4B_E B_D B_{Reconst}. \tag{58}$$

This completes the proof for the encoder gradient difference bound. $\qquad\square$

### G.2    DECODER GRADIENT DIFFERENCE UPPER BOUND

For decoder, assume that:

1. **Decoder Lipschitz Continuity:**

$$\|D^{(t+1)} - D^{(t)}\| \leq L_D \|\Theta_{dec}^{(t+1)} - \Theta_{dec}^{(t)}\|. \tag{59}$$

2. **Decoder Gradient Smoothness:**

$$\Big\|\nabla D^{(t+1)} - \nabla D^{(t)}\Big\| \leq \beta_D \|\Theta_{dec}^{(t+1)} - \Theta_{dec}^{(t)}\|. \tag{60}$$

   For simpility, we also assume $\beta_D$-smooth with respect to its input which helps to keep the proof concise:

$$\Big\|\nabla D(f_1; \Theta_{dec}) - \nabla D(f_2; \Theta_{dec})\Big\| \leq \beta_D \big\|f_1 - f_2\big\|. \tag{61}$$

3. **Boundedness:** There exist constants $B_D$ and $B_{Reconst}$ such that

$$\|\nabla D^{(t+1)}\big(E(\cdot; \Theta_{enc}^{(t+1)})\big)\| \leq B_D, \tag{62}$$

   and

$$\Big\|D^{(t+1)}\big(E(\cdot; \Theta_{enc}^{(t+1)})\big) - f\Big\| \leq B_{Reconst}. \tag{63}$$

4. **Encoder Influence:** The encoder is $L_E$-Lipschitz with respect to its parameters; that is,

$$\|E(\cdot; \Theta_{enc}^{(t+1)}) - E(\cdot; \Theta_{enc}^{(t)})\| \leq L_E \|\Theta_{enc}^{(t+1)} - \Theta_{enc}^{(t)}\|. \tag{64}$$

Then the gradient difference with respect to the decoder parameters satisfies

$$\|\nabla\mathcal{L}_{\text{E-D}}(\Theta_{dec}^{(t+1)}) - \nabla\mathcal{L}_{E-D}(\Theta_{dec}^{(t)})\| \leq 2B_{Reconst}\,\beta_D\,L_E\,\|\Theta_{enc}^{(t+1)} - \Theta_{enc}^{(t)}\| \tag{65}$$
$$+ 2B_{Reconst}\beta_D\|\Theta_{dec}^{(t+1)} - \Theta_{dec}^{(t)}\| + 4B_D B_{Reconst}$$

*Proof.* We begin with the gradient with respect to the decoder parameters at iteration $t$, so that

$$\nabla\mathcal{L}_{\text{E-D}}(\Theta_{dec}^{(t)}) = 2\Big[D^{(t)}\big(E(\cdot; \Theta_{enc}^{(t)})\big) - f\Big]\nabla D^{(t)}\big(E(\cdot; \Theta_{enc}^{(t)})\big). \tag{66}$$

Similarly, at iteration $t+1$,

$$\nabla\mathcal{L}_{\text{E-D}}(\Theta_{dec}^{(t+1)}) = 2\Big[D^{(t+1)}\big(E(\cdot; \Theta_{enc}^{(t+1)})\big) - f\Big]\nabla D^{(t+1)}\big(E(\cdot; \Theta_{enc}^{(t+1)})\big). \tag{67}$$

Define the difference:

$$\Delta_g = \Big\|\nabla\mathcal{L}_{\text{E-D}}(\Theta_{dec}^{(t+1)}) - \nabla\mathcal{L}_{E-D}(\Theta_{dec}^{(t)})\Big\|. \tag{68}$$

Thus,

$$\Delta_g = \Big\|2\Big[D^{(t+1)}\big(E(\cdot; \Theta_{enc}^{(t+1)})\big) - f\Big]\nabla D^{(t+1)}\big(E(\cdot; \Theta_{enc}^{(t+1)})\big)$$
$$- 2\Big[D^{(t)}\big(E(\cdot; \Theta_{enc}^{(t)})\big) - f\Big]\nabla D^{(t)}\big(E(\cdot; \Theta_{enc}^{(t)})\big)\Big\|. \tag{69}$$

To proceed, we add and subtract the intermediate term

$$2\Big[D^{(t+1)}\big(E(\cdot;\Theta_{enc}^{(t+1)})\big) - f\Big]\nabla D^{(t+1)}\big(E(\cdot;\Theta_{enc}^{(t)})\big) \tag{70}$$

to obtain:

$$\begin{aligned}\Delta_g = \Big\| &2\Big[D^{(t+1)}\big(E(\cdot;\Theta_{enc}^{(t+1)})\big) - f\Big]\Big(\nabla D^{(t+1)}\big(E(\cdot;\Theta_{enc}^{(t+1)})\big) - \nabla D^{(t+1)}\big(E(\cdot;\Theta_{enc}^{(t)})\big)\Big) \\ &+ 2\Big(\big[D^{(t+1)}\big(E(\cdot;\Theta_{enc}^{(t+1)})\big) - f\big]\Big)\Big(\nabla D^{(t+1)}\big(E(\cdot;\Theta_{enc}^{(t)})\big) - \nabla D^{(t)}\big(E(\cdot;\Theta_{enc}^{(t)})\big)\Big\| \\ &+ 2\Big(\big[D^{(t+1)}\big(E(\cdot;\Theta_{enc}^{(t+1)})\big) - f\big] - \big[D^{(t)}\big(E(\cdot;\Theta_{enc}^{(t)})\big) - f\big]\Big)\nabla D^{(t)}\big(E(\cdot;\Theta_{enc}^{(t)})\big)\Big\|.\end{aligned} \tag{71}$$

Applying the triangle inequality, we have:

$$\Delta_g \leq T_A + T_B + T_c, \tag{72}$$

where

$$T_A = 2\Big\|\big[D^{(t+1)}\big(E(\cdot;\Theta_{enc}^{(t+1)})\big) - f\big]\Big(\nabla D^{(t+1)}\big(E(\cdot;\Theta_{enc}^{(t+1)})\big) - \nabla D^{(t+1)}\big(E(\cdot;\Theta_{enc}^{(t)})\big)\Big)\Big\|, \tag{73}$$

and

$$T_B = 2\Big\|\Big(\big[D^{(t+1)}\big(E(\cdot;\Theta_{enc}^{(t+1)})\big) - f\big]\Big)\Big(\nabla D^{(t+1)}\big(E(\cdot;\Theta_{enc}^{(t)})\big) - \nabla D^{(t)}\big(E(\cdot;\Theta_{enc}^{(t)})\big)\Big\|. \tag{74}$$

and

$$T_C = 2\Big(\big[D^{(t+1)}\big(E(\cdot;\Theta_{enc}^{(t+1)})\big) - f\big] - \big[D^{(t)}\big(E(\cdot;\Theta_{enc}^{(t)})\big) - f\big]\Big)\nabla D^{(t)}\big(E(\cdot;\Theta_{enc}^{(t)})\big)\Big\| \tag{75}$$

**Bounding $T_A$:** Using the decoder gradient bound, we have

$$\Big\|\nabla D^{(t+1)}\big(E(\cdot;\Theta_{enc}^{(t+1)})\big) - \nabla D^{(t+1)}\big(E(\cdot;\Theta_{enc}^{(t)})\big)\Big\| \leq \beta_D\Big\|E(\cdot;\Theta_{enc}^{(t+1)}) - E(\cdot;\Theta_{enc}^{(t)})\Big\|. \tag{76}$$

By the encoder Lipschitz property,

$$\Big\|E(\cdot;\Theta_{enc}^{(t+1)}) - E(\cdot;\Theta_{enc}^{(t)})\Big\| \leq L_E\|\Theta_{enc}^{(t+1)} - \Theta_{enc}^{(t)}\|. \tag{77}$$

Also, by the reconstruction error bound,

$$\Big\|D^{(t+1)}\big(E(\cdot;\Theta_{enc}^{(t+1)})\big) - f\Big\| \leq B_{Reconst}. \tag{78}$$

Therefore,

$$T_A \leq 2B_{Reconst}\,\beta_D\,L_E\,\|\Theta_{enc}^{(t+1)} - \Theta_{enc}^{(t)}\|. \tag{79}$$

**Bounding $T_B$:** For $T_B$, we have

$$\Big\|\nabla D^{(t+1)}\big(E(\cdot;\Theta^{(t)}enc)\big) - \nabla D^{(t)}\big(E(\cdot;\Theta^{(t)}enc)\big)\Big\| \leq \beta_D\Big\|\Theta_{dec}^{(t+1)} - \Theta_{dec}^{(t)}\Big\| \tag{80}$$

Since:

$$\Big\|D^{(t+1)}\big(E(\cdot;\Theta^{(t+1)}enc)\big) - f\Big\| \leq B_{Reconst} \tag{81}$$

Thus, it follows that

$$T_B \leq 2B_{Reconst}\beta_D\|\Theta_{dec}^{(t+1)} - \Theta_{dec}^{(t)}\| \tag{82}$$

**Bounding $T_C$:** We have:

$$\Big\|\big[D^{(t+1)}\big(E(\cdot;\Theta^{(t+1)}enc)\big) - f\big] - \big[D^{(t)}\big(E(\cdot;\Theta^{(t)}enc)\big) - f\big]\Big\| \leq 2B_{Reconst} \tag{83}$$

and

$$\|\nabla D^{(t+1)}\big(E(\cdot;\Theta_{enc}^{(t+1)})\big)\| \le B_D, \tag{84}$$

Thus:

$$T_C \le 4B_D B_{Reconst} \tag{85}$$

**Combining $T_A$, $T_B$ and $T_C$:** We then have:

$$\begin{aligned}
\Delta_g &\le T_A + T_B + T_C \\
&\le 2B_{Reconst}\,\beta_D\,L_E\,\|\Theta_{enc}^{(t+1)} - \Theta_{enc}^{(t)}\| + 2B_{Reconst}\beta_D\|\Theta_{dec}^{(t+1)} - \Theta_{dec}^{(t)}\| + 4B_D B_{Reconst}
\end{aligned} \tag{86}$$

This completes the proof for the decoder-side gradient difference bound. $\square$

## H  PROOF OF PROXY TASK LOSS BOUNDS

**Theorem 3.** *Proxy Task Loss Bounds under a Lipschitz-dependent assumption between the masked graph signal and the raw graph signal, $\|\bar{f} - f\| \le \delta$. For our HarmonyGNNs ,*

$$\|S(\bar{f};\Phi) - T(f;\Psi)\| \le L_E \cdot \delta + \epsilon_{T-S}. \tag{87}$$

*For the encoder-decoder models,*

$$\|D\big(E(\bar{f};\Phi_{enc});\Theta_{dec}\big) - f\| \le L_E \cdot L_D \cdot \delta + \epsilon_{E-D}. \tag{88}$$

*W.L.O.G., assume $\epsilon_{E-D} = \epsilon_{T-S}$, our HarmonyGNNs has a smaller error upper bound, indicating that our teacher–student model is closer to the optimal solution $\Phi^*$ during training, which in turn implies that its parameter updates are more stable and its convergence speed is faster (as shown in the first result above).*

*Proof.*

$$\begin{aligned}
\|D\big(E(\bar{f};\Phi_{enc});\Theta_{dec}\big) - f\| &\le \|f - D\big(E(f;\Phi_{enc});\Theta_{dec}\big)\| \tag{89} \\
&\quad + \|D\big(E(f;\Phi_{enc});\Theta_{dec}\big) - D\big(E(\bar{f};\Phi_{enc});\Theta_{dec}\big)\| \\
&\le \epsilon_{E-D} + L_E L_D \|f - \bar{f}\| \tag{90} \\
&\le \epsilon_{E-D} + L_E \cdot L_D \cdot \delta \tag{91}
\end{aligned}$$

$$\begin{aligned}
\big\|S(\bar{f};\Phi) - T(f;\Psi)\big\| &\le \big\|S(\bar{f};\Phi) - S(f;\Phi)\big\| + \big\|S(f;\Phi) - T(f;\Psi)\big\| \tag{92} \\
&\le L_E \big\|\bar{f} - f\big\| + \epsilon_{T-S} \tag{93} \\
&\le L_E \delta + \epsilon_{T-S} \tag{94}
\end{aligned}$$

$$\square$$

## I  PROOF OF LINEAR CONVERGENCE BOUNDS

### I.1  ENCODER-DECODER:

**Theorem 4.** *Linear Convergence Bounds Under Strong Convexity. For our HarmonyGNNs ,*

$$\|\Phi^{(t+1)} - \Phi^*\|^2 \le \left(1 - \frac{\mu_E^2}{\beta_E^2}\right) \cdot \|\Phi^{(t)} - \Phi^*\|^2 \tag{95}$$

*For the encoder-decoder models,*

$$\|\theta^{(t+1)} - \theta^*\|^2 \leq \left(1 - \frac{\min(\mu_E^2, \mu_D^2)}{\max(\beta_E^2, \beta_D^2)}\right) \|\theta^{(t)} - \theta^*\|^2 \tag{96}$$

*from which we can see our HarmonyGNNs converges to the optimal solution $\Phi^*$ faster than the encoder-decoder counterpart to their optimal solutions $\Theta^*$ due to a smaller contraction factor $\left(1 - \frac{\mu_E^2}{\beta_E^2}\right) < \left(1 - \frac{\min(\mu_E^2, \mu_D^2)}{\max(\beta_E^2, \beta_D^2)}\right)$. This implies that HarmonyGNNs can achieve a faster convergence.*

*Proof.* From above, We can get the smoothness assumptions:

$$\|\nabla E(\cdot; \Theta_{enc}^{(t+1)}) - \nabla E(\cdot; \Theta_{enc}^{(t)})\| \leq \beta_E \|\Theta_{enc}^{(t+1)} - \Theta_{enc}^{(t)}\|. \tag{97}$$

and

$$\left\|\nabla D^{(t+1)} - \nabla D^{(t)}\right\| \leq \beta_D \|\Theta_{dec}^{(t+1)} - \Theta_{dec}^{(t)}\|. \tag{98}$$

Besides, we also assume strong convexity:

1. $\mu_E$-strong convexity of encoder:

$$\langle \nabla E(\bar{f}; \Theta_{enc}^{(t+1)}) - \nabla E(\bar{f}; \Theta_{enc}^{(t)}), \Theta_{enc}^{(t+1)} - \Theta_{enc}^{(t)}\rangle \geq \mu_E \cdot \|\Theta_{enc}^{(t+1)} - \Theta_{enc}^{(t)}\|^2 \tag{99}$$

2. $\mu_D$-strong convexity of decoder:

$$\langle \nabla D(\bar{f}; \Theta_{dec}^{(t+1)}) - \nabla D(\bar{f}; \Theta_{dec}^{(t)}), \Theta_{dec}^{(t+1)} - \Theta_{dec}^{(t)}\rangle \geq \mu_D \cdot \|\Theta_{dec}^{(t+1)} - \Theta_{dec}^{(t)}\|^2 \tag{100}$$

When combining an encoder and decoder, the overall strong convexity constant is often at most $\min(\mu_E, \mu_D)$ in a conservative sense.

Then for the encoder–decoder model, we define $\theta = (\Theta_{enc}, \Theta_{dec})$ for simplicity, where $\theta$ is used as a generic parameter vector for the entire model. The gradient descent update is given by:

$$\theta_{t+1} = \theta_t - \eta \nabla L_{ED}(\theta_t) \tag{101}$$

Following the gradient analysis:

$$\|\theta_{t+1} - \theta^*\|^2 = \|(\theta_t - \eta \nabla L_{ED}(\theta_t)) - \theta^*\|^2 \tag{102}$$

$$= \|\theta_t - \theta^*\|^2 - 2\eta \langle \nabla L_{ED}(\theta_t), \theta_t - \theta^*\rangle + \eta^2 \|\nabla L_{ED}(\theta_t)\|^2 \tag{103}$$

For $\langle \nabla L_{ED}(\theta_t), \theta_t - \theta^*\rangle$:

Since $\mu$-**strongly convex**, the following inequality holds:

$$L(\theta') \geq L(\theta) + \nabla L(\theta)^\top (\theta' - \theta) + \frac{\mu}{2}\|\theta' - \theta\|^2. \tag{104}$$

Let $\theta^*$ denote the global optimum of $L(\theta)$, i.e.,

$$\theta^* = \arg\min_\theta L(\theta). \tag{105}$$

then:

$$\nabla L(\theta^*) = 0. \tag{106}$$

Substituting $\theta' = \theta^*$ into the strong convexity definition, we obtain:

$$L(\theta^*) \geq L(\theta_t) + \nabla L(\theta_t)^\top (\theta^* - \theta_t) + \frac{\mu}{2}\|\theta^* - \theta_t\|^2. \tag{107}$$

Rearranging the terms, we have:

$$L(\theta^*) - L(\theta_t) \geq \nabla L(\theta_t)^\top (\theta^* - \theta_t) + \frac{\mu}{2} \|\theta^* - \theta_t\|^2. \tag{108}$$

Since $\theta^*$ is the global minimum, it follows that $L(\theta^*) \leq L(\theta_t)$. Therefore:

$$L(\theta^*) - L(\theta_t) \leq 0. \tag{109}$$

Combining the two inequalities:

$$0 \geq \nabla L(\theta_t)^\top (\theta^* - \theta_t) + \frac{\mu}{2} \|\theta^* - \theta_t\|^2. \tag{110}$$

$$\nabla L(\theta_t)^\top (\theta_t - \theta^*) \geq \frac{\mu}{2} \|\theta_t - \theta^*\|^2. \tag{111}$$

In general, the encoder and decoder are each $\mu_E$-strongly convex and $\mu_D$-strongly convex with respect to their parameters, respectively, then the composition can only guarantee a smaller strong convexity coefficient $\min(\mu_E, \mu_D)$ in the worst case, then:

$$\langle \nabla L_{ED}(\theta_t), \theta_t - \theta^* \rangle \geq \min(\mu_E, \mu_D) \|\theta_t - \theta^*\|^2$$

Similarly, for $\|\nabla L_{ED}(\theta_t)\|^2$, since $\nabla L_{ED}(\theta^*) = 0$, then

$$\|\nabla L_{ED}(\theta)\| = \|\nabla L_{ED}(\theta) - \nabla L_{ED}(\theta^*)\| \leq \beta \|\theta - \theta^*\|. \tag{112}$$
$$\|\nabla L_{ED}(\theta)\|^2 \leq \beta^2 \|\theta - \theta^*\|^2. \tag{113}$$

In Encoder-Decoder, we have two sets of parameters $(\Theta_{enc}, \Theta_{dec})$ and we typically argue that

$$L_{ED}(\theta) \text{ is at most } (\beta_E\text{-smooth}) \times (\beta_D\text{-smooth}), \tag{114}$$

For simplicity, let $L_{ED}(\theta)$ is $\max(\beta_E, \beta_D)$-smooth:

$$\|\nabla L_{ED}(\theta_t)\|^2 \leq \max(\beta_E^2, \beta_D^2) \|\theta_t - \theta^*\|^2$$

Then we can get:

$$\|\theta_{t+1} - \theta^*\|^2 \leq (1 - 2\eta \min(\mu_E, \mu_D) + \eta^2 \max(\beta_E^2, \beta_D^2)) \|\theta_t - \theta^*\|^2 \tag{115}$$

We want to find the minimum of $(1 - 2\eta \min(\mu_E, \mu_D) + \eta^2 \max(\beta_E^2, \beta_D^2))$:

$$-2 \min(\mu_E, \mu_D) + 2\eta \max(\beta_E^2, \beta_D^2) = 0 \tag{116}$$

$$\eta = \frac{\min(\mu_E, \mu_D)}{\max(\beta_E^2, \beta_D^2)} \tag{117}$$

With optimal learning rate $\eta = \frac{\min(\mu_E, \mu_D)}{\max(\beta_E, \beta_D)}$, we obtain:

$$\|\theta_{t+1} - \theta^*\|^2 \leq (1 - \frac{\min(\mu_E^2, \mu_D^2)}{\max(\beta_E^2, \beta_D^2)}) \|\theta_t - \theta^*\|^2 \tag{118}$$

### I.2 HARMONYGNNs :

For our method, analyzing one step:

$$\|\Phi_{t+1} - \Phi^*\|^2 = \|(\Phi_t - \tilde{\eta}\nabla L_{TS}(\Phi_t)) - \Phi^*\|^2 \tag{119}$$

Similarly as above, with optimal learning rate $\tilde{\eta} = \mu_E/\beta_E$:

$$\|\Phi_{t+1} - \Phi^*\|^2 \leq (1 - \frac{\mu_E^2}{\beta_E^2})\|\Phi_t - \Phi^*\|^2 \tag{120}$$

Clearly, our proposed method achieves better convergence because:

$$\frac{\mu_E^2}{\beta_E^2} > \frac{\min(\mu_E^2, \mu_D^2)}{\max(\beta_E^2, \beta_D^2)} \tag{121}$$

This inequality holds because:

1. $\mu_E^2 \geq \min(\mu_E^2, \mu_D^2)$
2. $\beta_E^2 \leq \max(\beta_E^2, \beta_D^2)$

Obviously, our model yields a faster convergence rate. $\square$

## J PERFORMANCE PLOT

In this section, we present a radar plot to illustrate the advantages of our proposed HarmonyGNNs compared to major baselines across all datasets as shown in Figure 4. This figure clearly demonstrates our model's effectiveness.

## K WEIGHTED GCN VERSUS VANILLA GCN

In this section, we compare our proposed Weighted GCN (WGCN) against the standard GCN as low-pass filters for capturing homophilic patterns in graphs. Specifically, we evaluate both models across different numbers of layers $\ell, \ell'$ and hidden-dimension sizes $h$ on datasets of varying scale—Cornell, Actor, and Roman-Empire—and report the results in Table 12.

From the results, we can see that when the dimension of the model is smaller, WGCN consistently outperforms vanilla GCN. This is because WGCN can adapt message passing flexibly, assigning higher weights to similar neighbors while downweighting dissimilar ones in heterophilic regions. However, when heavier models are used, GCN can achieve comparable and even better performance than WGCN. We conclude that this is because the deeper GCN has sufficient learning capacity, whereas the larger number of learnable parameters in WGCN potentially causing overfitting. Additionally, we observe that WGCN still provides advantages when the graph has complex mixed patterns, such as in the Actor and Roman-Empire datasets.

In summary, when computational resources are limited and graphs exhibit complex structures, WGCN can learn better representations. These findings prove the effectiveness of our proposed WGCN approach.

## L OVERALL MASKING RATIO $R$

In this section, we provide an analysis of the overall masking ratio $R$, which determines the total percentage of nodes being masked for the student model during training. We present the results in Table 13. From the results, we can observe that different datasets require different optimal masking ratios, which is consistent with our conclusions in the main text. For datasets with more complicated patterns, such as Actor and Roman Empire, a smaller masking ratio proves beneficial. This prevents excessive node masking, which would otherwise prevent the student model from effectively capturing the teacher model's representations.

**Performance Comparison Across All Datasets**

Figure 4: Performance comparison across all datasets

Table 12: The effects of WGCN over Vanilla GCN

|  | Cornell | | Actor | | Roman-Empire | |
| --- | --- | --- | --- | --- | --- | --- |
|  | WGCN | GCN | WGCN | GCN | WGCN | GCN |
| $\ell=1, \ell'=2, h=32$ | 84.86±2.48 | 83.78±2.71 | 37.00±0.91 | 36.67±0.78 | 73.36±0.41 | 72.83±0.34 |
| $\ell=2, \ell'=3, h=32$ | 85.03±2.00 | 84.32±2.31 | 37.23±0.77 | 36.95±0.95 | 74.02±0.38 | 73.85±0.57 |
| $\ell=1, \ell'=2, h=256$ | 85.40±1.79 | **85.68**±2.11 | 37.80±0.56 | 37.83±0.75 | 74.32±0.48 | 74.64±0.56 |
| $\ell=2, \ell'=3, h=256$ | 85.21±1.89 | 85.21±2.01 | **38.15**±0.71 | 38.10±0.53 | **75.86**±0.47 | 75.60±0.57 |

## M    ENCODED TOKEN SELECTION STRATEGIES

In this section, we present a study on the token selection strategies mentioned in our main text. Specifically, we evaluate four strategies:

- Directly selecting the first token $X_{0,C}$
- Taking the mean across all tokens
- Taking the maximum across all tokens
- Performing hierarchical token fusion as described in Eqn. 7

Our results demonstrate that the proposed hierarchical token combination performs best among all strategies when dealing with large, complex graphs. This is because it can combine the similar encoded tokens first and dynamically learns their combination weights in a coarse-to-fine manner, which demonstrates the effectiveness of this design. Simply selecting the first encoded token results in significant information loss and performs worse than basic aggregation methods like mean and

Table 13: The effects of the overall masking ratio $R$ (Eqn. 4)

| Ratio $R$ | Cornell | Actor | Roman-Empire |
|---|---|---|---|
| 1 | 84.98±3.01 | 36.88±0.98 | 73.65±0.47 |
| 0.8 | **85.68**±2.11 | 37.32±0.66 | 74.40±0.42 |
| 0.5 | 85.26±2.25 | **38.15**±0.71 | **75.86**±0.47 |
| 0.3 | 84.86±1.93 | 37.53±0.56 | 75.32±0.34 |

max pooling, as evidenced in the Roman Empire dataset. However, for smaller datasets, simpler selection methods are sufficient since hierarchical learning can potentially cause overfitting.

In our proposed method, the selection of these strategies is treated as a hyperparameter that can be easily adjusted based on the specific properties of the dataset. This flexibility highlights the adaptability of our model design to different graph scenarios.

Table 14: The effects of different token selection strategies (Eqn. 7)

| | Cornell | Actor | Roman-Empire |
|---|---|---|---|
| First Token | **85.68**±2.11 | 37.00±0.82 | 72.46±0.57 |
| Mean | 85.32±2.53 | 37.30±0.72 | 75.12±0.40 |
| Max | 85.26±2.88 | 37.56±0.88 | 74.87±0.79 |
| Hierarchical | 84.98±2.22 | **38.15**±0.71 | **75.86**±0.47 |

# N   HETEROPHILY AND HOMOPHILY IN GRAPHS

## N.1   DATASETS DESCRIPTIONS

We provide a basic introduction of heterophilic datasets used in our experiments (Pei et al., 2020; Platonov et al., 2023) and present T-SNE visualizations of four representative examples—Cornell, Texas, Wisconsin, and Actor—to illustrate their complex mixing patterns.

**WebKB**. The WebKB1 dataset is a collection of web pages. Cornell, Wisconsin and Texas are three sub-datasets of it. Nodes represent web pages and edges denote hyperlinks between them. The node features are bag-of-words representations of the web pages, which are manually categorized into five classes: student, project, course, staff, and faculty.

**Actor Co-occurrence Network**. This dataset is derived from the film-director-actor-writer network. In this network, each node corresponds to an actor, and an edge between two nodes indicates that the actors co-occur on the same Wikipedia page. The node features consist of keywords extracted from these Wikipedia pages, and the actors are classified into five categories based on the content of their pages.

**Roman-Empire**. The Roman-empire dataset is built from the full text of the English Wikipedia article on the Roman Empire (=22.7K words). Each word is a node, with edges connecting words that are adjacent in the text or linked by a dependency relation. Nodes are labeled by their part-of-speech roles (17 most frequent plus "other"), and node features are 300-dimensional fastText embeddings. The resulting graph is extremely sparse and chain-like (avg. degree =2.9, diameter =6,824) and exhibits strong heterophily, making it a challenging benchmark for GNNs to capture long-range and syntactic dependencies.

**Wikipedia Network**. Chameleon and squirrel are two page-page networks on specific topics in Wikipedia. Nodes represent web pages and edges represent mutual links between pages. Node features correspond to informative nouns appearing in the Wikipedia pages. These datasets are used for node classification tasks, where pages are classified into five categories based on their average monthly traffic.

Upon closer examination, researchers (Platonov et al., 2023) identified a critical flaw in these widely-used benchmark datasets: a substantial portion of nodes are duplicates with identical regression

targets and neighborhood structures. In the squirrel dataset, 57% of nodes (2,978 out of 5,201) are duplicates, while in chameleon, duplicates account for 61% of nodes (1,387 out of 2,277). These duplicates create problematic train-test data leakage, as they appear across training, validation, and testing splits.

To remedy this issue, researchers developed filtered versions by removing nodes that had no incoming edges and shared both the same monthly traffic value and outgoing edge set with another node in the graph. Testing on these filtered datasets revealed dramatically different results - many models that performed exceptionally well on the original datasets showed significant performance degradation, and the relative rankings of different models changed substantially. This finding suggests that previous evaluations based on the original datasets were unreliable, as models may have been exploiting data leakage rather than learning meaningful graph patterns.

## N.2 PATTERN ANALYSIS

**Wisconsin, Texas and Cornell**: These three datasets are relatively small and exhibit high heterophily. In the raw feature visualizations (left), nodes of different labels are highly mixed, with significant overlap between categories. After applying HarmonyGNNs , the right-side visualizations reveal a more distinct clustering structure, where nodes of the same label are more compactly grouped. For instance, in Texas and Cornell, purple nodes appear more concentrated, and red nodes are better distinguished from other categories, indicating that the model effectively captures the structural patterns. In Wisconsin, the node clusters become more distinguishable, with clearer boundaries between different categories. This demonstrates the model's ability to learn meaningful representations that enhance classification and clustering tasks.

**Actor**: This dataset contains a large number of nodes with an imbalanced label distribution (with red nodes being dominant). In the raw feature space (left), although red nodes are mainly centered, other colored nodes remain scattered without clear boundaries. Notably, the outer ring of nodes effectively represents the mixed structural pattern, which accounts for the relatively low accuracy observed in both node classification and node clustering tasks across all models. In the HarmonyGNNs embedding space (right), red nodes are more tightly clustered, while nodes of other labels form relatively well-separated subclusters. This suggests that the model improves class separation and enhances discrimination among different node categories.

Overall, these visualizations demonstrate that in the HarmonyGNNs embedding space, nodes of different categories form more distinguishable clusters compared to the raw feature space. This intuitively explains why our model achieves great performance in both node classification and node clustering tasks. Furthermore, it highlights the model's strong representation learning capability across various graph structures, whether homophilic or heterophilic.

## O DATASETS STATISTICS

We provide the deatils of datasets used in our experiment here. The homophily ratio, denoted as homo, represents the proportion of edges that connect two nodes within the same class out of all edges in the graph. Consequently, graphs with a strong homophily ratio close to 1, whereas those with a ratio near 0 exhibit strong heterophily.

$$\text{homo} = \frac{\left|\{(u,v) \in E \mid y_u = y_v\}\right|}{|E|} \tag{122}$$

## P HYPERPARAMETERS

Our model's hyperparameters are tuned from the following search space:

- Learning rate for SSL model: $\{0.01, 0.005, 0.001\}$.
- Learning rate for classifier: $\{0.01, 0.005, 0.001\}$.

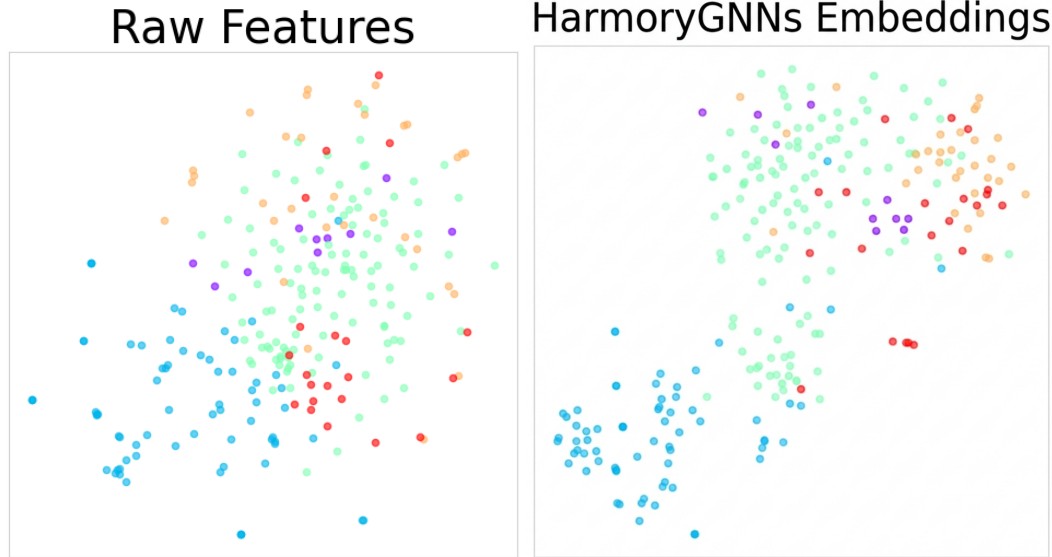

Figure 5: T-SNE visualizations of Wisconsin datasets.

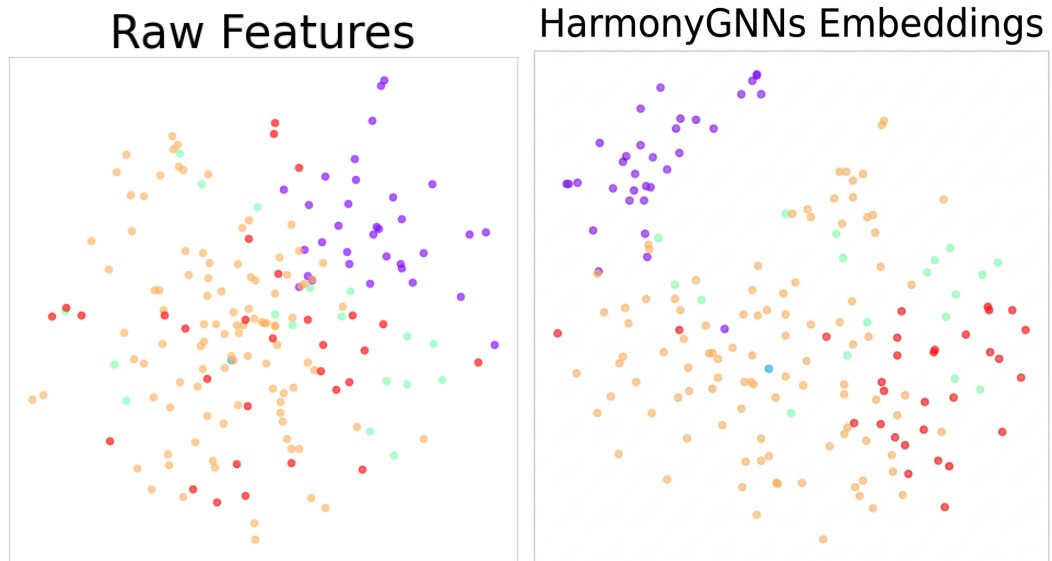

Figure 6: T-SNE visualizations of Texas datasets.

- Weight decay for SSL model: $\{0,\ 1 \times 10^{-3},\ 5 \times 10^{-3},\ 8 \times 10^{-3},\ 1 \times 10^{-4},\ 5 \times 10^{-4},\ 8 \times 10^{-4}\}$.
- Weight decay for classifier: $\{0,\ 5 \times 10^{-4},\ 5 \times 10^{-5}\}$.
- Dropout for Filters: $\{0.1,\ 0.3,\ 0.5,\ 0.7,\ 0.8\}$.
- Dropout for Attention: $\{0.1,\ 0.3,\ 0.5,\ 0.7,\ 0.8\}$.
- Dimension of tokens: $\{128,\ 256,\ 512,\ 1024,\ 2048,\ 4096\}$.
- Hidden units of filters: $\{16,\ 32,\ 64,\ 128,\ 256,\ 512\}$.
- Total masking ratio: $\{0.9,\ 0.8,\ 0.5,\ 0.3,\ 0.1,\ 0\}$.
- Dynamic masking ratio: $\{0.9,\ 0.8,\ 0.5,\ 0.3,\ 0.1,\ 0\}$.

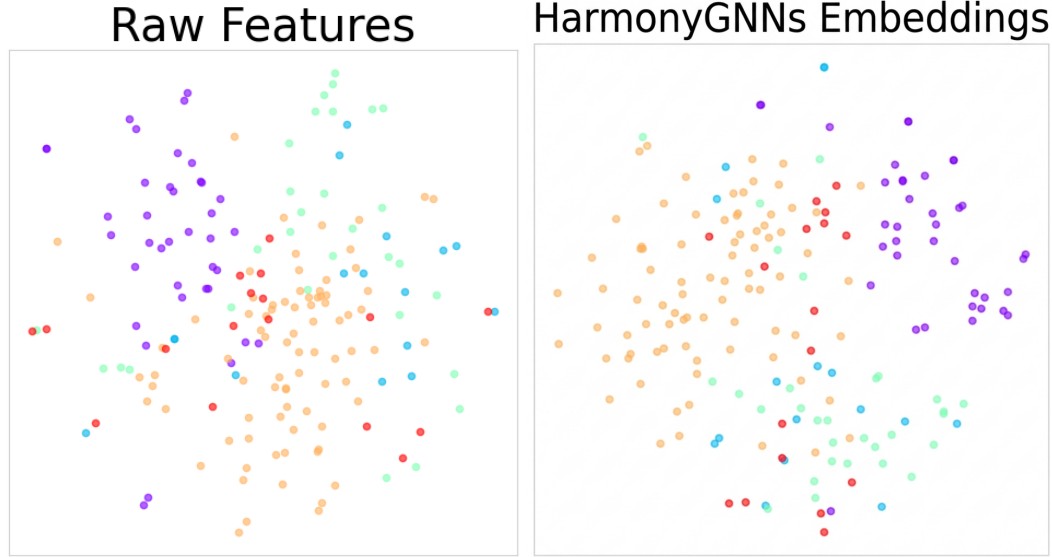

Figure 7: T-SNE visualizations of Cornell datasets.

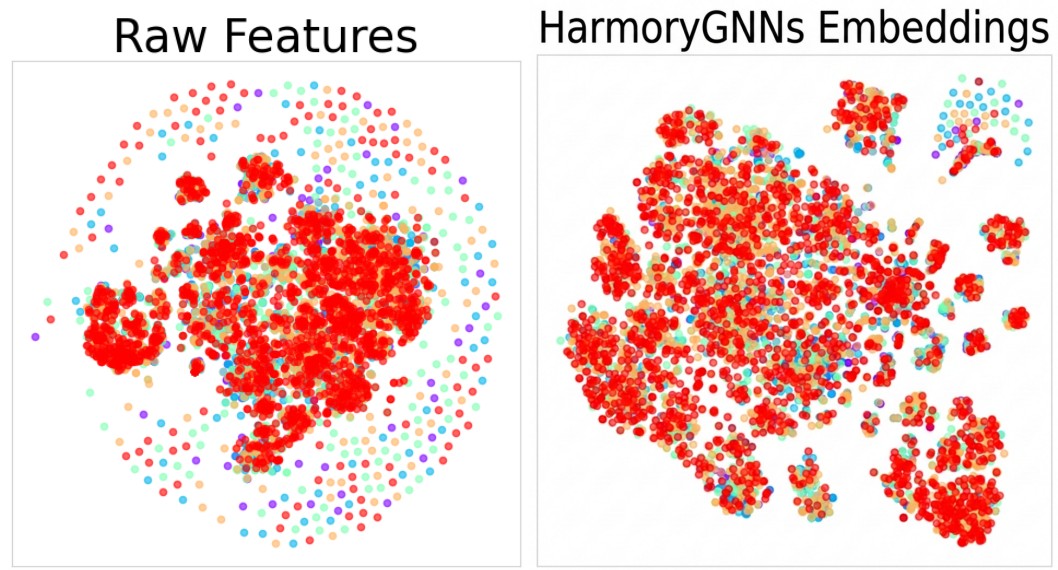

Figure 8: T-SNE visualizations of Actor datasets.

- Momentum: $\{0.9,\ 0.99,\ 0.999\}$.

## Q  THE USE OF LARGE LANGUAGE MODELS (LLMs)

We used Large Language Models (LLMs) solely to refine the writing.

Table 15: Datasets statistics.

| Datasets | Node | Edges | Feats | Classes | Homo |
|---|---|---|---|---|---|
| Cornell | 183 | 295 | 1,703 | 5 | 0.30 |
| Texas | 183 | 309 | 1,703 | 5 | 0.11 |
| Wisconsin | 251 | 499 | 1,703 | 5 | 0.21 |
| Actor | 7,600 | 29,926 | 932 | 5 | 0.22 |
| Chameleon(Filtered) | 890 | 17708 | 2325 | 5 | 0.24 |
| Squirrel(Filtered) | 2223 | 93996 | 2089 | 5 | 0.21 |
| Roman-Empire | 22662 | 32927 | 300 | 18 | 0.05 |
| Cora | 2708 | 10,556 | 1,433 | 7 | 0.81 |
| CiteSeer | 3,327 | 9,104 | 3,703 | 6 | 0.74 |
| PubMed | 19,717 | 88,648 | 500 | 3 | 0.80 |
| Ogbn-Arxiv | 169343 | 1166243 | 128 | 40 | 0.66 |

