# OpenReview forum: "HarmonyGNNs: Harmonizing Heterophily and Homophily in GNNs via Self-Supervised Node Encoding"
_ICLR.cc/2026/Conference — ICLR 2026 Poster_

### Official Review · Reviewer_Gv2M · 2025-10-28

[review text omitted: it was posted to a different submission]

---

> ### Author Response · Authors · 2025-11-12
> **Kind Request for Review Clarification**
>
> Dear Reviewer Gv2M,
>
> Thank you so much for the efforts of reviewing our submission. We noticed that the comments you posted appear to reference a different paper, as the content does not correspond to ours. We respectfully think that either this was caused by a potential system
> error, or that this have been posted mistakenly.
>
> We would appreciate it if you could kindly verify this and provide feedback specific to our paper.  We would greatly value your expert assessment of our work and remain committed to addressing any concerns you may identify.
>
> Thank you again for your time and effort in the review process!
>
> Best regards,
>
> Authors

---

> > ### Comment · Reviewer_Gv2M · 2025-11-12
> >
> > Dear Authors,
> >
> > Thanks for a quick reminder on this error. I apologize for the confusion caused. I have updated the reviews and let me know if there are still any other issues you discovered.
> >
> > Best regards,

---

> ### Author Response · Authors · 2025-11-21
>
> Thanks for your updated review and support. We are happy to address your concerns.
>
> ## Response to Weakness 1
>
> Thanks for your suggestion. Given the tight time of the rebuttal, we will re-organize the writing of the whole introduction according to your advice in our final revision. And for line 93-96, we explain it in details as follows:
>
> Existing SSL methods struggle to handle mixed patterns for two reasons. First, contrastive approaches (e.g., InfoNCE) rely on relative instance discrimination without a global reference. In mixed graphs, this creates ambiguity, as the model cannot adaptively determine whether to aggregate or distinguish neighbors, leading to inconsistent local criteria instead of a single, coherent latent space. Second, generative methods typically force the reconstruction of raw features from neighbors. In heterophilic regions, where neighbors have dissimilar attributes, this objective creates contradictory supervision signals, hindering effective learning.
>
>
> ## Response to Weakness 2 and Question 1
>
> We provide these definitions here and will include them in the revised version to improve readability.
>
> **node-difficulty**
>
> Formally, we define the node-difficulty score after each training iteration as Eq. (2) in the submission:
>
> $\mathrm{Diffi}(v) =|| S(v) - T(v) ||_2^2$
>
> where $T(v)$ is the teacher's embedding of node $v$ (obtained from the full, unmasked graph) and
> $S(v)$ is the student's embedding (obtained from the masked graph).
> Thus, node-difficulty is precisely the squared prediction error of the student with respect to
> the teacher in the latent space.
>
> Intuitively, $\mathrm{Diffi}(v)$ measures how hard node $v$ currently is for the student:
>
> - If $\mathrm{Diffi}(v)$ is small, the student already matches the teacher well on $v$, so masking
>   $v$ again yields limited benefit.
>
> - If $\mathrm{Diffi}(v)$ is large, the student still fails to reconstruct the teacher's representation
>   of $v$, meaning that $v$ lies in a *difficult region* (e.g., structurally ambiguous).
>
> Therefore, the node-difficulty score is a key component of our objective harmonization, as it determines how the proxy task is defined within our SSL framework.
>
> **exploitation ratio**
>
> First, we would like to point out that the name originally comes from reinforcement learning:
>
> - Exploitation means repeatedly choosing the action with the highest estimated reward, to maximally use what the agent already knows.
>
> - Exploration means occasionally trying other actions, to discover new information and avoid getting stuck in a suboptimal choice.
>
> In our case:
>
> - Exploitation corresponds to "Difficulty-Aware Masking": We "exploit" the model's current feedback (i.e., difficulty score) to specifically target nodes that are currently hard to learn. This maximizes the information gain of each training step.
>
> - Exploration corresponds to "Random Masking": We randomly select nodes to mask. This ensures easy or moderate nodes are not completely ignored and the learning process remains diverse.
>
> The exploitation ratio is a single hyperparameter that controls the exploration–exploitation trade-off in our masking strategy: it specifies what fraction of the masking budget is allocated to high-difficulty (hard) nodes versus uniformly random nodes. This allows the model to systematically focus more on informative hard nodes, while still maintaining sufficient randomness for coverage and avoiding over-fitting to a small subset of nodes.
>
> Specifically, the exploitation ratio $r \in [0,1]$ simply controls this balance:
> $r=0$ corresponds to pure random masking (pure exploration),
> $r=1$ corresponds to always masking the hardest nodes (pure exploitation),
> and intermediate $r$ values interpolate between them. Pease refer to Table 5 for detailed results.

---

> ### Author Response · Authors · 2025-11-21
> **Continued Response — Page 2**
>
> ## Response to Weakness 3
>
> First, **we emphasize that the goal of Sec.4.3—and of this paper more broadly—is not to claim orders-of-magnitude speedups, but to verify that our theoretically motivated design does not introduce significant additional overhead compared with SOTA baselines, while still achieving substantial accuracy gains.** On the challenging and large-scale mixed-pattern graphs Actor and Roman-Empire, H$^3$GNNs achieves strictly better performance with comparable or lower peak memory usage and non-trivial reductions in training time to convergence (see Table 3).
>
> Second, the theoretical analysis in Sec. 3.3 concerns the optimization behavior and conditioning of our predictive objective compared to encoder–decoder SSL frameworks. **It is not intended to guarantee a fixed percentage reduction in runtime on a specific GPU or dataset. In other words, it provides a theoretical foundation and guarantee that H$^3$GNNs can be both effective and efficient.** Our empirical results validate this point—while significantly improving performance, H$^3$GNNs also reduces or maintains overall training efficiency. We therefore do not view there as being an imbalance between theory and empirical effectiveness.
>
> Besides, for performance improvement, we believe that Tables 1, 2, 8, 10, and 11 clearly demonstrate the advantages of our method across diverse benchmarks.
>
> ## Response to Question 2
>
> Following your suggestion, we are happy to provide theoretical efficiency analysis.
>
> Let $N$ denote the number of nodes, $|E|$ the number of edges, $D$ the input feature dimension, and $H$ the hidden (channel) dimension.
> Let $K$ be the number of WGCN layers and $C$ the number of per-node tokens used in the harmonization module (a small constant, e.g., $C = 4$ in our implementation).
>
> **Computational complexity**
> The computation of H³GNNs can be decomposed into three parts:
>
> - **MLP branch.**
>   Mapping raw node features to node-specific tokens via one or two linear/MLP layers costs $\mathcal{O}(N D H)$ or $\mathcal{O}(N D H + N H^2)$ (for a 2-layer MLP).
>
> - **WGCN branch.**
>   The $K$-hop structural embeddings are implemented with $K$ sparse WGCN layers, each with complexity $\mathcal{O}(|E| H)$.
>   Hence the cost is $\mathcal{O}(K |E| H)$.
>
> - **Feature-level MHSA.**
>   For each node, we apply MHSA over its $C$ feature tokens. Computing attention scores between $C$ tokens and the associated linear projections costs $\mathcal{O}\big(C^2 H + CH^2\big)$ per node, and thus $\mathcal{O}(N C^2 H + N C H^2)$ over all nodes.
>   **Note that the cost is not $\mathcal{O}(N^2 H)$. This is exactly why we emphasize feature-level attention instead of node-wise attention.**
>
> Putting everything together, the overall complexity of the student encoder is $\mathcal{O}\big(N D H + K |E| H + N C^2 H + N C H^2\big)$.  Since $K$ and $C$ are small constants (e.g., $K \le 2$ and $C = 4$) in our implementation, this simplifies to  $\mathcal{O}\big(N D H + |E| H + N H^2\big)$, which is on the same order as standard GNNs and strictly more efficient than node-wise graph transformers with $\mathcal{O}(N^2 H)$ attention.

---

> ### Author Response · Authors · 2025-11-26
> **A Friendly Reminder**
>
> Dear Reviewer Gv2M,
>
> Thanks again for your valuable comments and support. As the rebuttal deadline is approaching, we would like to kindly follow up to confirm whether our responses have addressed all of your concerns.
>
> We deeply value your feedback and remain committed to improving our work based on your expert guidance.
>
> Best regards,
>
> Authors

---

> > ### Comment · Reviewer_Gv2M · 2025-11-27
> >
> > Thanks for your detailed responses. I think my concerns are largely addressed based on the promised revision from the authors. I wish to keep my positive score.

---

> > > ### Author Response · Authors · 2025-12-03
> > >
> > > Dear Reviewer Gv2M.
> > >
> > > Thank you very much. We appreciate your recognition of our submission and rebuttal.
> > >
> > > Thanks

---

### Official Review · Reviewer_RngT · 2025-10-30

**Soundness:** 3
**Presentation:** 3
**Contribution:** 3
**Rating:** 8
**Confidence:** 3

**Summary:**

The paper proposes H3GNNs, a self-supervised framework aimed at unifying the learning of homophilic and heterophilic graphs within a single model. It introduces a teacher–student predictive architecture with dynamic node-difficulty-based masking to create informative self-supervision signals, and a joint structural node encoding module that fuses linear, nonlinear, and structural features through a Weighted GCN and Transformer-based hierarchical fusion. Experiments on multiple benchmark datasets show that H3GNNs outperforms prior self-supervised GNN models.

**Strengths:**

1.The paper shows reasonable originality by combining teacher–student predictive learning, dynamic masking, and weighted structural encoding to address both homophily and heterophily in self-supervised GNNs.
2.The technical quality is solid, with extensive experiments and consistent gains across benchmarks, though some design choices and analyses lack depth.
3.The paper is clearly written overall.

**Weaknesses:**

1.The roles of WGCN, MLP, linear projection, and transformer fusion are not clearly separated or analyzed; it is unclear how each module contributes to handling homophily and heterophily individually.
2.The dynamic masking update relies on previous loss values, yet the paper does not specify how frequently these scores are recomputed or how the warm-up and exploitation phases are scheduled.
3.The efficiency comparison table does not include information about the hardware platform, GPU type, or framework version, so the claimed improvements in training time and memory cannot be verified.
4.The paper lacks systematic sensitivity analysis for critical hyperparameters like the number of WGCN layers.
5.On homophilic datasets, the reported improvements are minor and may fall within variance, yet no statistical tests or confidence intervals are presented to confirm significance.
6.The theoretical analysis assumes strong convexity and smoothness for deep GNNs, which are unrealistic; hence the derived convergence results have limited practical meaning.

**Questions:**

1.Could the authors provide a clearer explanation or visualization of how the WGCN, MLP, linear projection, and transformer fusion interact? It is not obvious which module primarily contributes to handling heterophily versus homophily.
2.Why are exactly four “tokens” used in the joint encoding, and why is the fusion order fixed? Have the authors tried using different token numbers or fusion sequences, and how sensitive is performance to these choices?
3.How are the learnable edge weights in WGCN regularized or constrained to prevent degenerate solutions (e.g., trivial scaling or sparsity collapse)? Are they shared across layers or trained independently?
4.The dynamic masking variants show small numerical differences. Can the authors provide more analysis (e.g., convergence behavior, difficulty distribution, or qualitative examples) to demonstrate why dynamic masking is preferable to random masking?

---

> ### Author Response · Authors · 2025-11-21
>
> We greatly appreciate your comments and support, and we provide detailed responses to all of your questions below.
>
> ## Response to Weakness 1 and Question 1
>
> To clarify, WGCN, MLP, and linear projection serve as distinct  components in our graph encoder, each specifically designed to capture different structural patterns (homophily or heterophily). No single component alone is sufficient to model all types of patterns effectively (see Section 2 and Figure 3 for details). The transformer block plays a key role in representation harmonization, as it performs effective fusion of these complementary representations—an essential step when homophilic and heterophilic patterns coexist within complex graphs.
>
> Next, we would like to show how each component contributes and how they interact in details.
>
> **High-level design principle.**
> Representation harmonization is built upon a Joint Structural Node Encoding mechanism, which enables seamless learning across regions exhibiting different structural patterns. To achieve this, we first design specialized encoder modules that target distinct types of graph patterns, and then employ a Multi-Head Self-Attention (MHSA) block to jointly harmonize their encoded representations into a unified embedding space.
>
> Specifically, each component plays a distinct role:
>
> - **Specificity Branch --- MLP & linear projection**: node-specific representations (approximated heterophily-awareness);
>
> - **Structure Branch --- WGCN**: structure-aware representations (homophily-awareness but preserve heterophily),
>
> - **Harmonization Module --- Transformer**: feature level harmonization between representations from two branches above.
>
> **Linear projection $f^{(Linear)}(v)$ and MLP $f^{(Mlp)}(v)$: heterophily-oriented node specificity.**
>
> As discussed in Section 2, classical GNNs often fail on highly heterophilic graphs, while a simple MLP performs sufficiently well under strongly heterophilic settings. This observation motivates us to include a branch for handling heterophily:
>
> - The linear projection can map the raw feature sapce into latent space and play crucial roles when neighborhoods exhibit high heterophily.
>
> - The MLP adds nonlinear capacity to model complex feature patterns.
>
> **WGCN $H^{(l)}$: homophily-oriented but heterophily-adaptive structural awareness.**
>
> Similarly, as discussed in Section 2, we observe that in homophilic graphs (or locally homophilic regions of complex graphs), GNNs still demonstrate strong performance due to their ability to encode structural information via message passing. This makes them naturally aligned with homophilic patterns. Moreover, WGCN enhances traditional GNNs by learning edge weights on top of a normalized adjacency initialization, enabling the model to downweight dissimilar neighbors in heterophilic regions. This design is especially beneficial in cases where a node's neighbors exhibit complex and mixed structural patterns.
>
> **Transformer fusion: feature-level harmonization between $f^{(Linear)}(v)$, $f^{(Mlp)}(v)$ and $H^{(l)}$**
>
> The key of representation harmonization is the proposed feature-level attention mechanism that takes the encoded representation in above branches and **dynamically decides, for each node, how much to trust and balance specificity branch and structure branch**, The subsequent hierarchical fusion then merges these embeddings according to their semantic roles. **Please refer to the red-highlighted block in Figure 1 for a visualization of their interaction.**
>
> Ablation results in Table 4 and 9 show clear and consistent drops when remove the transformer, demonstrating that the harmonization module is the primary reason our model can adaptively handle both homophily and heterophily.
>
> ## Response to Weakness 2
> Since computing the difficulty score incurs no additional cost, we update it after every epoch following a warm-up phase. The number of warm-up epochs is treated as a hyperparameter; however, as it does not significantly impact performance, we fixed it to 20 in our implementation and didn't specify in the submission due to the space limit.
>
> ## Response to Weakness 3
> All experiments were conducted on a single NVIDIA A100 GPU with 80 GB memory, using PyTorch 2.6.0 and PyG 2.6.1.
>
> ## Response to Weakness 4
> In our submission, we have provide such an ablation study on the number of layers and hidden dimensions of WGCN, along with a comparison to vanilla GCN, in **Appendix K (Table 12)**. The results show that even with only a few layers (1-2 layers), WGCN achieves strong performance. Please refer to the appendix for detailed analysis, and other additional studies such as overall masking ratio and token fusion strategies.

---

> ### Author Response · Authors · 2025-11-21
> **Continued Response — Page 2**
>
> ## Response to Weakness 5
>
> Following your suggestion, we conducted statistical significance testing on two homophilic graphs where our model achieves the best performance, Cora and ogbn-arxiv. We compare H³GNNs against GraphACL and GraphMAE using the 10 official splits, which are the strongest baselines on these datasets. Using a commonly adopted significance level of $\alpha = 0.05$, we found that all $p$-values are significantly smaller than $\alpha$, demonstrating the statistical significance of our improvements.
>
> | Model     | Cora      | Ogbn-Arxiv |
> |-----------|-----------|------------|
> | GraphACL  | 2.68e-08  | 1.42e-03   |
> | GraphMAE  | 4.18e-08  | 2.94e-05   |
>
> ## Response to Weakness 6
>
> First, the goal of Section 3.3 is not to provide a fully general, global theory for arbitrary non-convex deep GNNs, which would be unrealistic. Instead, we clarify that we analyze a simplified and commonly used setting where the training dynamics can be approximated by a strongly convex and smooth objective (e.g., in a local neighborhood).
>
> Hence, we want to state that these are **relative** statements between two SSL formulations under same assumption, rather than claims that the full deep GNN training landscape is globally convex. Thus, while the network is globally non-convex, the assumptions are appropriate for deriving **comparative** insights.
>
> Next, we also want to mention that assumptions of strong convexity and smoothness are standard in optimization and in much of the existing theory for deep learning models [1, 2, 3]. They universally adopt similar simplifications to make the analysis mathematically tractable.
>
> Finally, our theoretical results are not used in isolation. The analysis is used to explain and justify observed empirical results, not to claim that real-world GNN training is exactly strongly convex. It provides meaningful theoretical insight into why H$^3$GNNs train more stably and effectively in practice, rather than being of limited practical value.
>
> [1] GNNAutoScale: Scalable and Expressive Graph Neural Networks via Historical Embeddings
>
> [2] Understanding Self-Supervised Learning Dynamics without Contrastive Pairs
>
> [3] A Convergence Theory for Deep Learning via Over-Parameterization
>
> ## Response to Question 2
>
> We clarify that the number of tokens and the fusion strategy are principled designs derived from the goal of "Harmonization," rather than arbitrary choices.
>
> **Why exactly four tokens?**
>
> The number of tokens is a hyperparameter that can be easily tuned. As stated in our response to A1, the current selection is intentionally designed to cover both homophilic and heterophilic patterns commonly found in real-world graphs. Specifically, $f^{(\mathrm{Linear})}(v)$ and $f^{(\mathrm{MLP})}(v)$ make the model heterophily-aware by capturing both linear and non-linear feature-level relationships, while $H^{(\ell)}(v)$ and $H^{(\ell')}(v)$ enable homophily-awareness by encoding both local and long-range neighbor information. Thus, the current choice of four tokens is *not* an arbitrary hyperparameter, but rather the most effective configuration for capturing both homophily and heterophily across most datasets.
>
> We would also like to emphasize the flexibility of our design: both the number and the types of tokens are tunable hyperparameters that can be readily adapted to different datasets (e.g., by adding/removing tokens or stacking additional layers per token). We keep the current configuration fixed across datasets because it already performs well and ensures implementation simplicity.
>
> **Fusion Strategy** Regarding the fusion strategies, we provide an ablation study in **Appendix M** that compares different combinations (e.g., mean, max). In particular, for hierarchical fusion, the ordering is conceptually motivated by semantics rather than arbitrary.
>
> We first fuse the two "homophiliy-aware" tokens $H^{(\ell)}(v), H^{(\ell')}(v)$ to obtain a representation, and then hierarchically merge this summary with the two "heterophily-aware" tokens. This reflects the intended semantics: first aggregate multi-scale structure into a coherent "homophiliy-aware" representation, then align it with node-specific "heterophily-aware" tokens to obtain the final harmonized embedding $S(v)$.
>
> We indeed experimented with alternative strategies, such as reversing the fusion order (i.e., fusing heterophily-aware tokens first, followed by homophily-aware ones). However, these alternative strategies resulted in inferior performance, and therefore were not included in the submission.

---

> ### Author Response · Authors · 2025-11-21
> **Continued Response — Page 3**
>
> ## Response to Question 3
>
> **preventing degenerate solution**
> First, as defined in Eq.~4), the element $\mathcal{A}_{ij}$ is a learnable edge weight initialized from the normalized adjacency
> $\tilde{A} = \tilde{D}^{-1/2}(A + I)\tilde{D}^{-1/2}$.
> In implementation, we do not learn a dense $N \times N$ matrix: the learnable parameters are stored in sparse form and only for edges present in $(A+I)$.
> Thus, the graph topology is fixed and the sparsity pattern of $A$ coincides with the original graph.
>
> Second, in our implementation, the learned edge weight are passed through a Sigmoid function:
> $a_e = \sigma(w_e) \in (0,1)$
> as the edge weight for edge $e$.
> Thus, the actual propagation matrix uses *sigmoid-bounded* weights: edge weights are strictly between $0$ and $1$, cannot grow unbounded, and cannot change sign.
> This removes the typical "trivial scaling" degree of freedom.
>
> Finally, we initialize $\mathcal{A}$ exactly at the normalized adjacency $\tilde{A}$, so that at the beginning of training WGCN reduces to a standard GCN-style propagation with well-conditioned scale.
> The optimizer therefore starts from a non-degenerate, empirically strong baseline and only fine-tunes from the normalized adjacency matrix, rather than searching over arbitrary scaling factors from scratch.
>
> **Parameter sharing across layers.**
> As written in Eq.~(4), $\mathcal{A}$ does not carry a layer index; thus the same edge-weight matrix $\mathcal{A}$ is shared. It could also help reduces the degrees of freedom and the complexity.
>
> Please also refer the ablation study comparing WGCN with vanilla GCN in Appendix K in our submission for more details.
>
> ## Response to Question 4
>
> Random masking treats all nodes as equally informative. In contrast, our dynamic strategies explicitly exploit the teacher--student discrepancy which measures how difficult node $v$ is for the student to predict given the teacher. Thus, the SSL objective allocates more capacity to structurally ambiguous where random masking is most likely to waste supervision. Please refer to Table 4, 5, 9 for comparison.

---

> ### Author Response · Authors · 2025-11-26
> **A Friendly Reminder**
>
> Dear Reviewer RngT,
>
> Thanks again for your valuable comments and support. As the rebuttal deadline is approaching, we would like to kindly follow up to confirm whether our responses have addressed all of your concerns.
>
> We deeply value your feedback and remain committed to improving our work based on your expert guidance.
>
> Best regards,
>
> Authors

---

### Official Review · Reviewer_6ntd · 2025-11-01

**Soundness:** 4
**Presentation:** 3
**Contribution:** 3
**Rating:** 6
**Confidence:** 3

**Summary:**

The paper introduces a unified SSL framework addressing the challenge of learning on graphs with mixed structural patterns. It achieves this through Representation Harmonization (joint structural node encoding with WGCN and self-attention) and Objective Harmonization (a predictive teacher–student architecture with dynamic masking). The method effectively balances homophilic and heterophilic signals, showing strong performance across benchmarks. Overall, H3GNNs offers a significant advancement in adaptive, structure-aware graph self-supervised learning.

**Strengths:**

The strength of this paper lies in its innovative unification of homophily and heterophily modeling within a self-supervised graph learning framework. The authors identify a key limitation in existing GNN and SSL methods—the inability to handle mixed structural patterns—and propose a comprehensive solution (H3GNNs) that achieves both representation harmonization and objective harmonization. It provides both stability and adaptability in learning from complex graph structures. Moreover, the paper is well-written, with extensive experiments showing state-of-the-art performance on heterophilic and mixed graphs, demonstrating strong generalization and interpretability.

**Weaknesses:**

The paper lacks experiments on large-scale graphs and comparisons with more recent or relevant algorithms, limiting its scalability claims. Efficiency analysis is narrow, involving few baselines without clear justification for their selection as representative methods. Broader comparisons and rationale would strengthen the empirical evaluation and conclusions.

**Questions:**

See weakness

---

> ### Author Response · Authors · 2025-11-21
>
> We thank the reviewer for the thoughtful comments and support, and we are happy to provide detailed answers to your concerns.
>
> ## Response to Weakness
>
> First, we want to clarify that **the primary objective of our paper and the selected baselines are to enhance model learning effectiveness on complex graphs with mixed patterns without sacrificing efficiency, rather than to pursue scalability improvements.** And our dataset selection is consistent with all baseline methods to ensure fair comparison.
>
> Second, regarding the efficiency analysis in Table 3, the two heterophilic graphs (Actor and Roman-Empire) are considered large-scale benchmarks[1]—they are also used by baseline methods to analyze efficiency at scale.
>
> Regarding baseline selection, as mentioned in our submission, we chose MUSE and GREET because: (1) they achieve strong performance with publicly available hyperparameter settings, enabling reproducible comparisons (e.g., recent baseline S3GCL does not publish code); and (2) they employ SSL-based training and are specifically designed for learning on heterophilic graphs (see Section 4 for detailed dataset and baseline selection rationale). **Additionally, we provide performance comparisons with recent baselines using their reported results in Appendix F**, which we believe sufficiently demonstrates our model's advantages.
>
> Following your suggestion, we are happy to provide additional efficiency analysis through both theoretical and empirical perspectives.
>
> **Theoretical complexity analysis**
>
> Let $N$ denote the number of nodes, $|E|$ the number of edges, $D$ the input feature dimension, and $H$ the hidden (channel) dimension.
> Let $K$ be the number of WGCN layers and $C$ the number of per-node tokens used in the harmonization module (a small constant, e.g., $C = 4$ in our implementation).
>
> **Computational complexity.**
> The computation of H³GNNs can be decomposed into three parts:
>
> - **MLP branch.**
>   Mapping raw node features to node-specific tokens via one or two linear/MLP layers costs
>   $\mathcal{O}(N D H)$ or $\mathcal{O}(N D H + N H^2)$ (for a 2-layer MLP).
>
> - **WGCN branch.**
>   The $K$-hop structural embeddings are implemented with $K$ sparse WGCN layers, each with complexity
>   $\mathcal{O}(|E| H)$.
>   Hence the cost is
>   $\mathcal{O}(K |E| H)$.
>
> - **Feature-level MHSA.**
>   For each node, we apply MHSA over its $C$ feature tokens.
>   Computing attention scores between $C$ tokens and the associated linear projections costs
>   $\mathcal{O}\big(C^2 H + C H^2\big)$ per node,
>   and thus $\mathcal{O}(N C^2 H + N C H^2)$ over all nodes. **Note that the cost is not $\mathcal{O}(N^2 H)$. This is exactly why we emphasize feature-level attention instead of node-wise attention.**
>
> Putting everything together, the overall complexity of the student encoder is
> $\mathcal{O}\big(N D H + K |E| H + N C^2 H + N C H^2\big)$. Since $K$ and $C$ are small constants (e.g., $K \le 2$ and $C = 4$) in our implementation, this simplifies to
> $\mathcal{O}\big(N D H + |E| H + N H^2\big)$,
> which is on the same order as standard GNNs and strictly more efficient than node-wise graph transformers with $\mathcal{O}(N^2 H)$ attention.
>
> **Memory complexity.**
> The memory footprint of the proposed method is dominated by (i) node activations, (ii) the sparse graph structure, and (iii) model parameters:
>
> - **Node activations.**
>   Each node stores $C$ tokens of dimension $H$, leading to
>   $\mathcal{O}(N C H) = \mathcal{O}(N H)$
>
> - **Graph structure.**
>   The sparse edge index and a scalar weight per edge in WGCN requires
>   $\mathcal{O}(|E|)$
>
> - **Model parameters.**
>   Input projections, WGCN weights, and token-level attention/MLPs take $\mathcal{O}(D H + H^2)$, which is **independent** of $N$ and $|E|$.
>
> Under the common sparse-graph assumption, the total memory complexity is therefore
> $\mathcal{O}(N H + |E| + D H + H^2)$,
> i.e., it grows **linearly** in both $N$ and $|E|$ and does not require storing any $\mathcal{O}(N^2)$ node-wise attention maps.
>
> **Further Empirical efficiency analysis**
>
> To ensure consistency and fair comparison with baseline methods. we add additional efficiency analysis on ogbn-arxiv (the largest homophilic benchmark used in baselines) below:
>
> | Model   | Memory(MB) | Epoch Time (s) | Total Time(s) |
> |---------|--------|------------|-------------|
> | H³GNNs  | 33486  | 4.03       | 476.16      |
> | MUSE    | 38791  | 5.31       | 627.32      |
> | GREET   | 35096  | 6.71       | 738.86      |
>
> [1] S3gcl: Spectral, swift, spatial graph contrastive learning.

---

> ### Author Response · Authors · 2025-11-26
> **A Friendly Reminder**
>
> Dear Reviewer 6ntd,
>
> Thanks again for your valuable comments and support. As the rebuttal deadline is approaching, we would like to kindly follow up to confirm whether our responses have addressed all of your concerns.
>
> We deeply value your feedback and remain committed to improving our work based on your expert guidance.
>
> Best regards,
>
> Authors

---

### Official Review · Reviewer_VdwQ · 2025-11-01

**Soundness:** 3
**Presentation:** 2
**Contribution:** 2
**Rating:** 4
**Confidence:** 4

**Summary:**

This paper proposes a unified SSL framework for both homophilic and heterophilic graph. The framework is built on a teacher-student predictive architecture, where the student network achieves representation harmonization via joint structural node encoding. The experimental results show the effectiveness of the proposed framework.

**Strengths:**

1.	The idea of : teacher–student predictive architecture is interesting, which eliminate the need for complex negative sampling.

2.	The author provides theoretical analysis of the proposed method, although the analysis is really complicated and hard to understand.

3.	The proposed method outperforms existing SSL methods on heterophilic graph datasets.

**Weaknesses:**

1. Besides the teacher-student predictive architecture, the overall novelty is limited and the design of the student network is somewhat engineering (e.g., Learning Multi-Head Self-Attention and Fusing and Selecting Tokens Hierarchically as SSL Node Encoding). The motivation of these strategies is unclear.

2. The overall framework involves huge memory and computation overhead. It is suggested to analyze the memory complexity and computation complexity in detail. The authors perform computation and memory comparisons in Section 4.3. It would be better to perform similar experiments on more datasets (i.e., homophilic graphs and large-scale graphs).

3. Many details are missing. For example, in Section 3.1, the authors claimed that the teacher network is not trained. It is not clear how to get the parameters of the teacher network. Moreover, the architecture of the teacher network is also not introduced.

**Questions:**

see Weaknesses.

---

> ### Author Response · Authors · 2025-11-21
>
> Thank you for your comments. We provide the following clarification addressing your concerns.
>
> ## Response to Weakness 1
>
>  With all due respect, we would like to take this opportunity to clarify that our design motivation and contributions may have been potentially misunderstood, and the comment does not accurately reflect the core technical advances in our work. **We would like to emphasize that the core novelty of H³GNNs does not lie in inventing isolated components from scratch, but rather in the novel synthesis and framework design to address a previously unsolved challenge: harmonizing heterophily and homophily within a single, unified graph SSL framework at both the objective level and the representation level.**
> To the best of our knowledge, no prior work has achieved a unified model as our proposed H$^3$GNN  that consistently performs robustly across diverse graph structures—from highly homophilic graphs to highly heterophilic ones.
>
> Next, we elaborate on our technical novelties and contributions:
>
> As articulated in the introduction, **the first core novelty of H³GNNs lies in posing graph SSL with mixed patterns as a two-layer harmonization problem**:
>
> + **Objective harmonization**: how could th SSL proxy objective balance signals from homophilic and heterophilic regions?
> + **Representation harmonization**: how could the encoder preserve information across different graph patterns?
>
> To address this two-layer harmonization problem, our technical novelties include:
>
> + *Novel objective harmonization design beyond teacher–student SSL*: Beyond the teacher–student predictive architecture, our objective introduces two key innovations essential for harmonizing mix patterns (validated in Table 9 of the appendix):
>
>     - *All-node latent-space prediction:*
>      Unlike prior SSL methods that predict only masked nodes in the raw feature space, we introduce a new all-node prediction loss in the latent space (Eq. (1)): the student must match teacher embeddings for both masked and unmasked nodes. This design is specifically motivated by the connectivity in graph-structured tasks.
>
>     - *Two novel node-difficulty–driven masking curricula:*
>     These strategies create a dynamic training curriculum tailored to graph data, forcing the model to learn more meaningful representations by focusing on nodes that are structurally harder to predict. It is motivated by the fact that some regions in mixed-pattern graphs pose greater learning challenges than others (see Appendix N)
>
> + *Representation harmonization rather than being a generic stack of modules:*
>
>     - Our method applies MHSA at the **feature level** across different representations within each node by joint structural node encoding instead of at the node–node level as in graph transformers. This is the key design that enables a unified model to effectively learn structural information under mixed homophilic and heterophilic patterns, and it showcases **a novel application of the vanilla MHSA mechanism**.
>
>     - Our proposed WGCN is a well-tailored architectural design motivated by the modeling need to adaptively modulate the weight of homophilic and heterophilic neighbors during message passing, providing a principled way to encode both types of structural relationships.
>
>     - A novel hierarchical feature-token fusion method ensures that the final embedding captures multi-level structural semantics.
>
> **The other core novelty lies in the interplay between these two components, both of which are essential:**
>
> + Without representation harmonization, the T-S framework would be guiding a model that is "blind" to the different structural representations it needs to adapt to.
>
> + Without objective harmonization, our powerful encoder would lack a principled and stable learning objective, making it difficult to transform its awareness of different patterns into robust, generalizable representations.
>
>
>
> Please check Table 4 and 9 for more detailed ablation studies of these designs. Besides, we provide comprehensive theoretical analyses and empirical validation across diverse graphs to support our approach. In summary, H³GNNs introduces **a novel framework** that harmonizes heterophily and homophily through objective and representation levels. All components are tightly justified by both analysis and ablations, and we believe they represent substantial conceptual and technical novelty beyond baselines.

---

> ### Author Response · Authors · 2025-11-21
> **Continued Response — Page 2**
>
> ## Response to Weakness 2
>
> We appreciate the reviewer’s suggestion. However, the concerns raised stem from a misunderstanding. First, we would like to clarify that the computational costs reported in Table 3 are reasonable given the large scale of the two benchmarks (Actor and Roman-Empire), where our approach remains more efficient than existing state-of-the-art methods. Second, while achieving comparable or even better efficiency, our model significantly outperforms these baselines in terms of performance.
>
> As you suggested, we briefly summarize the computational and memory complexity of H³GNNs to relate the empirical efficiency results in Table 3.
>
> **Notation.**
> Let $N$ denote the number of nodes, $|E|$ the number of edges, $D$ the input feature dimension, and $H$ the hidden (channel) dimension.
> Let $K$ be the number of WGCN layers and $C$ the number of per-node tokens used in the harmonization module (a small constant, e.g., $C = 4$ in our implementation).
>
> **Computational complexity**
> The computation of H³GNNs can be decomposed into three parts:
>
> - **MLP branch.**
>   Mapping raw node features to node-specific tokens via one or two linear/MLP layers costs $\mathcal{O}(N D H)$ or $\mathcal{O}(N D H + N H^2)$ (for a 2-layer MLP).
>
> - **WGCN branch.**
>   The $K$-hop structural embeddings are implemented with $K$ sparse WGCN layers, each with complexity $\mathcal{O}(|E| H)$.
>   Hence the cost is $\mathcal{O}(K |E| H)$.
>
> - **Feature-level MHSA.**
>   For each node, we apply MHSA over its $C$ feature tokens. Computing attention scores between $C$ tokens and the associated linear projections costs $\mathcal{O}\big(C^2 H + CH^2\big)$ per node, and thus $\mathcal{O}(N C^2 H + N C H^2)$ over all nodes.
>   **Note that the cost is not $\mathcal{O}(N^2 H)$. This is exactly why we emphasize feature-level attention instead of node-wise attention.**
>
> Putting everything together, the overall complexity of the student encoder is $\mathcal{O}\big(N D H + K |E| H + N C^2 H + N C H^2\big)$.  Since $K$ and $C$ are small constants (e.g., $K \le 2$ and $C = 4$) in our implementation, this simplifies to  $\mathcal{O}\big(N D H + |E| H + N H^2\big)$, which is on the same order as standard GNNs and strictly more efficient than node-wise graph transformers with $\mathcal{O}(N^2 H)$ attention.
>
> ---
>
> **Memory complexity**
> The memory footprint of the proposed method is dominated by:
>
> - **Node activations.**
>   Each node stores $C$ tokens of dimension $H$, leading to $\mathcal{O}(N C H) = \mathcal{O}(N H)$
>
> - **Graph structure.**
>   The sparse edge index and a scalar weight per edge in WGCN require $\mathcal{O}(|E|)$
>
> - **Model parameters.**
>   Input projections, WGCN weights, and token-level attention/MLPs take $\mathcal{O}(D H + H^2)$, which is **independent** of $N$ and $|E|$.
>
> Under the common sparse-graph assumption, the total memory complexity is therefore  $\mathcal{O}(N H + |E| + D H + H^2)$,  i.e., it grows **linearly** in both $N$ and $|E|$ and does not require storing any $\mathcal{O}(N^2)$ node-wise attention maps.
>
> ---
>
> **large-scale datasets**
> Our evaluation focuses on the Actor and Roman-Empire datasets to ensure consistency and fair comparison with baseline methods, as these are the large-scale heterophilic graphs used in prior work [1]. Since additional large-scale datasets were not evaluated by the baselines, it is not feasible for us to compare against them without access to their best hyperparameter settings. Therefore, we follow the existing setup for efficiency analysis, and we believe the current results are sufficient to demonstrate the advantages of our model over the baselines.
>
> Following your suggestion, we provide additional efficiency analysis on ogbn-arxiv (the largest homophilic benchmark used in baselines) below:
>
> | Model   | Memory(MB) | Epoch Time (s) | Total Time(s) |
> |---------|--------|------------|-------------|
> | H³GNNs  | 33486  | 4.03       | 476.16      |
> | MUSE    | 38791  | 5.31       | 627.32      |
> | GREET   | 35096  | 6.71       | 738.86      |
>
> [1] S3GCL: Spectral, swift, spatial graph contrastive learning.

---

> ### Author Response · Authors · 2025-11-21
> **Continued Response — Page 3**
>
> ## Response to Weakness 3
>
> Due to space limitations, **we believe we described the design verbally in Section 3.1.** However, we are happy to provide more details here for your reference:
>
> **Training Strategy and Teacher Model Architecture.** Regarding the architecture, as clearly stated in Section 3.1, *"The teacher network has the exact same network configuration as the student,"* where the student's Joint Structural Node Encoding module is detailed in Section 3.2. In terms of training, we also mentioned in the same paragraph that the teacher model does not require independent training; its parameters are computed via exponential moving average (EMA) of the student model's parameters as:
>
> $$\Psi_{i} = \alpha \cdot \Psi_{i-1} + (1-\alpha)\cdot \Phi_i, \quad i=1,2,\cdots, I$$
>
> where $\alpha$ is tuned from $\{0.9, 0.99, 0.999\}$. EMA is a commonly used strategy in SSL to facilitate training stability.
>
> More specifically, this design provides several advantages: (1) it offers stable and consistent targets by providing a consistent semantic representation in the latent space, which is crucial when homophilic and heterophilic regions coexist, as the teacher observes the full graph; (2) it maintains low computational cost since it requires no independent training; and (3) its effectiveness is theoretically justified in Theorem 1.
>
> In summary, the teacher model in Section 3.1 is (i) architecturally well-defined (same encoder as the student), (ii) parametrically well-specified (EMA of the student without training), and (iii) functionally essential to our objective harmonization.

---

> ### Author Response · Authors · 2025-11-26
> **A Friendly Reminder**
>
> Dear Reviewer VdwQ,
>
> Thanks again for your valuable comments. As the rebuttal deadline is approaching, we would like to kindly follow up to confirm whether our responses have addressed all of your concerns.
>
> If there are any remaining questions or issues, we would welcome the opportunity to address them before the deadline. If you believe that our responses have satisfactorily resolved your concerns, we would greatly appreciate your consideration in updating the evaluation accordingly.
>
> We deeply value your feedback and remain committed to improving our work based on your expert guidance.
>
> Best regards,
>
> Authors

---

### Meta-Review · Area_Chair_vjwH · 2025-12-13

**Summary:**

Three of the four reviewers provided positive initial scores. During the rebuttal, only Reviewer Gv2M participated in the discussion, since his comments correspond to another paper. Considering that all reviewers who provided positive initial ratings have low confidence, I checked the paper by myself. I am inclined to agree with Reviewer VdwQ, whose concerns focus on the novelty. In summary, I agree to accept this paper by considering the reviewers’ comments,  but I wouldn't mind if the paper gets rejected.

**Reviewer Concerns:**

Reviewer Gv2M shows that his concerns were largely addressed. The other three reviewers didn’t participate in the discussion. Reviewer RngT mainly focuses on the clarity of the paper, and the authors provided clarification. Reviewer 6ntd concerns the computational complexity, and the authors provide convincing analysis and experiments. Reviewer VdwQ thinks the novelty is limited, and I believe the author's response is not convincing.

**Reviewer Scores:**

Only Reviewer Gv2M participated in the discussion, since his comments correspond to another paper. By checking the revision history, the rating of Reviewer Gv2M is Rating: 6 / Confidence: 2. Since most reviewers provide positive feedback, I believe they may maintain their scores as follows.
RngT	Rating: 8 / Confidence: 3
6ntd	        Rating: 6 / Confidence: 3
VdwQ	Rating: 4 / Confidence: 4
Gv2M	Rating: 6 / Confidence: 2  (refer to Revision)

---

### Decision · Program_Chairs · 2026-01-26

Accept (Poster)